# Farrerol directly activates the deubiquitinase UCHL3 to promote DNA repair and reprogramming when mediated by somatic cell nuclear transfer

Weina Zhang[1,2,6], Mingzhu Wang[1,3,6], Zhiwei Song[1,6], Qianzheng Fu[1,6], Jiayu Chen [1], Weitao Zhang[4], Shuai Gao[5], Xiaoxiang Sun[1], Guang Yang[1], Qiang Zhang[5], Jiaqing Yang[1], Huanyin Tang[1], Haiyan Wang[1], Xiaochen Kou[1], Hong Wang[1], Zhiyong Mao [1,2,4] ✉, Xiaojun Xu [4] ✉, Shaorong Gao [1] ✉ & Ying Jiang [1] ✉

Farrerol, a natural flavanone, promotes homologous recombination (HR) repair to improve genome-editing efficiency, but the specific protein that farrerol directly targets to regulate HR repair and the underlying molecular mechanisms have not been determined. Here, we find that the deubiquitinase UCHL3 is the direct target of farrerol. Mechanistically, farrerol enhanced the deubiquitinase activity of UCHL3 to promote RAD51 deubiquitination, thereby improving HR repair. Importantly, we find that embryos of somatic cell nuclear transfer (SCNT) exhibited defective HR repair, increased genomic instability and aneuploidy, and that the farrerol treatment post nuclear transfer enhances HR repair, restores transcriptional and epigenetic network, and promotes SCNT embryo development. Ablating UCHL3 significantly attenuates farrerol-mediated stimulation in HR and SCNT embryo development. In summary, we identify farrerol as an activator of the deubiquitinase UCHL3, highlighted the importance of HR and epigenetic changes in SCNT reprogramming and provide a feasible method to promote SCNT efficiency.

Efficient DNA repair preserves genetic information in cells; thus, DNA repair plays a basic and critical role in various biological processes, such as early embryonic development, aging, tumorigenesis and cellular reprogramming[1-3]. During these processes, in which cell division is necessary, the homologous recombination (HR) repair pathway is particularly important, as this pathway is responsible for repairing DNA double-strand breaks (DSBs) in the S/G2 phase; furthermore, defects in the HR pathway may result in embryonic lethality, early onset of aging, an increased incidence of tumorigenesis and defective cellular reprogramming[4-7]. Therefore, it is tempting to hypothesize

[1]Shanghai Key Laboratory of Maternal Fetal Medicine, Clinical and Translational Research Center of Shanghai First Maternity and Infant Hospital, Frontier Science Center for Stem Cell Research, School of Life Sciences and Technology, Tongji University, 200092 Shanghai, China. [2]Tsingtao Advanced Research Institute, Tongji University, 266071 Qingdao, China. [3]Jiaxing University Affiliated Women and Children Hospital, 314000 Jiaxing, China. [4]State Key Laboratory of Natural Medicines, China Pharmaceutical University, 210009 Nanjing, China. [5]Key Laboratory of Animal Genetics, Breeding and Reproduction of the MARA, National Engineering Laboratory for Animal Breeding, College of Animal Science and Technology, China Agricultural University, 100193 Beijing, China. [6]These authors contributed equally: Weina Zhang, Mingzhu Wang, Zhiwei Song, Qianzheng Fu. ✉e-mail: zhiyong_mao@tongji.edu.cn; xiaojunxu@cpu.edu.cn; gaoshaorong@tongji.edu.cn; ying_jiang@tongji.edu.cn

that targeting the repair factors that directly regulate the HR pathway might be a good approach to improve genomic stability and optimize these processes.

Somatic cell nuclear transfer (SCNT) is a well-characterized technology that helps somatic cells recapture totipotency, and the reprogrammed totipotent cells may further generate living individuals[8,9]. Since the efficiency of producing blastocysts and viable offspring by SCNT remains low[10,11], great efforts have been made to identify pivotal molecular factors to improve SCNT efficiency[12–14]. Previous studies from our and other laboratories indicated that combined epigenetic interventions could efficiently enhance the preimplantation development of SCNT embryos[15–17]. However, these interventions largely failed to promote the postimplantation development rate, suggesting that other than the epigenetic manipulation, there exist other unknown defects during the process of the postimplantation development of SCNT embryos. To complete SCNT-mediated reprogramming, the donor nuclei genome must be rapidly transitioned from a somatic to early embryonic state within 1–2 cell cycles to achieve zygotic genome activation (ZGA) and further development. The process involves the following major transitions: one to rapidly disaggregate the somatic-like 3D genome structure inherent in the donor cell chromatin while the other is that the donor nucleus undergoes a transition from a mitotic metaphase-like state to a meiotic metaphase II-like state[8,18]. As a comparison, the formation of primordial germ cells with a similar transformation process usually occurs over several weeks in mice[19]. Therefore, the nuclear genome of the donor cell during SCNT reprogramming is under enormous pressure and is extremely vulnerable to DNA damage[2,20]. Developing novel approaches that activate DNA repair to counteract the accumulated DNA damage during SCNT might improve the development potential of SCNT embryos.

Defects in HR repair destabilize genomes, resulting in the impairment in embryo development[21]. For instance, knocking out the recombinase RAD51, the rate-limiting HR factor, leads to preimplantational lethality[22], while disruption of RAD50 and BRCA1, two critical HR factors, reduces cell proliferation and causes abnormal embryonic development at the preimplantation stage[23,24]. Numerous E3 ligases and deubiquitinases participate in the regulation of HR repair by modifying key HR factors[25]. The deubiquitinase UCHL3 is a member of the ubiquitin carboxyl-terminal hydrolase (UCH) family[26], and it plays important roles in murine fertilization and preimplantation development[27]. In response to DNA damage, UCHL3 is phosphorylated and activated by ATM, and activated UCHL3 positively regulates HR by deubiquitinating RAD51 to promote the BRCA2-RAD51 interaction[28]. In addition, UCHL3 deubiquitinates and stabilizes TDP1 to promote the repair of TOP1-induced single-strand breaks[29], and UCHL3 also deubiquitinates KU80 to promote the retention of KU80 at DNA damage sites, thereby improving DNA repair by nonhomologous end joining (NHEJ)[30]. In addition to DNA repair factors, UCHL3 also targets a number of proteins that regulate other signaling pathways[31,32]. In various types of human cancers, UCHL3 is overexpressed, so several studies have tried to identify inhibitors of UCHL3 to develop potential cancer therapies[33–35]. Moreover, given the function of UCHL3 in upregulating HR, which is critical for maintaining genome integrity in normal somatic and stem cells, identifying the activators of UCHL3 could potentially improve the efficiency of cellular reprogramming and the quality of reprogrammed cells.

In our previous work, we identified that farrerol, a natural flavanone extracted from *Rhododendron dauricum L.*, promotes HR repair, thereby improving the efficiency of CRISPR–Cas9-mediated genome targeting[36]. In addition to its function in HR, farrerol exhibits other pharmacological activities[37–41]. For instance, farrerol attenuates Aβ-induced inflammation in BV-2 cells by activating the Nrf2/Keap1 pathway[37]. Farrerol induces the expression of two antioxidant enzymes, HO-1 and NQO1, by targeting GSK-3[38]. In addition, farrerol could potentially treat atherosclerosis and cancer and prevent bacterial infection[39–41]. Although prior research has shown that farrerol enhances RAD51 recruitment at DNA damage sites, the targets of farrerol that facilitate RAD51 recruitment and promote HR have not been discovered.

Here, using LiP-small molecule mapping (LiP-SMap) and surface plasmon resonance (SPR) assays, we identify UCHL3 as the direct target protein of farrerol, and the data revealed the strong affinity of farrerol for UCHL3 ($K_d$ = 36.73 nM). Further cocrystallization experiments, SPR assays and cellular thermal shift assays (CETSAs) demonstrate that the K187 and R215 residues in the UCHL3 protein are critical for the farrerol−UCHL3 association. Experiments on both in vitro and in vivo functions indicate that farrerol promotes the catalytic activity of UCHL3, thereby leading to a reduced ubiquitination level of RAD51 and enhanced HR repair. We further demonstrate that improving HR repair by farrerol is beneficial to SCNT-mediated cellular reprogramming, and that farrerol ameliorates the impaired transcriptome of SCNT embryos and contributes to proper lineage specification. Moreover, we find that farrerol could promote the developmental potential of SCNT embryos both in vitro and in vivo in a UCHL3-dependent manner, thus providing a simple and feasible method to improve SCNT efficiency.

## Results

### Identification of the deubiquitinase UCHL3 as a potential target of farrerol

Through screening a large number of small molecules in our previous work, we revealed that farrerol promoted the HR repair pathway without influencing NHEJ, and a preliminary mechanistic study demonstrated that farrerol promoted the recruitment of RAD51, but not RPA, at DNA damage sites[36]. To further dissect the regulatory mechanisms of farrerol on HR repair, we sought to determine the specific proteins that farrerol directly targets. We therefore performed a LiP-SMap assay to search for the cellular proteins to which farrerol exhibits a high affinity (Fig. 1a)[42]. The search revealed 101 different peptides that were potentially associated with farrerol (Supplementary Data 1). Further comparison between the control group and farrerol-treated group indicated that ten proteins were significantly changed (Fig. 1b), indicating that these proteins might be potential targets of farrerol.

Among these ten proteins, DNA-PKcs (PRKDC) and UCHL3 are involved in the regulation of DNA repair[28,43]. We then performed a CETSA assay to confirm this finding[44]. We found that incubating HEK293 cells with farrerol increased the thermal stability of UCHL3 but not DNA-PKcs (Fig. 1c, d and Supplementary Fig. 1a–c), and the generated UCHL3 thermal melting curves revealed that the value of Tm (the temperature at which 50% of proteins unfold and rapidly precipitate by heat) increased from 51.58 to 53.81 °C in farrerol-treated HEK293 cells (Fig. 1d). These data indicate that UCHL3, rather than DNA-PKcs, is likely the target of farrerol.

Next, to determine the affinity of farrerol to UCHL3, we performed SPR experiments using recombinant UCHL3 and farrerol at different concentrations. The results revealed a strong interaction between farrerol and UCHL3, and the dissociation constant (Kd) of farrerol/UCHL3 reached a low of 36.73 nM (Fig. 1e and Supplementary Fig. 1d).

### Farrerol binds to UCHL3 through two amino acid residues, Lys187 and Arg215

To elucidate how farrerol specifically binds to UCHL3, we crystallized UCHL3 in the presence of farrerol and determined the structure of the complex at 1.58 Å (Fig. 2a, b and Table 1). The phenolic hydroxyl of farrerol formed hydrogen bonds with two amino acid residues, Lys187 and Arg215, in UCHL3 (Fig. 2c). Next, we aligned the pre-deposited structure of UCHL3 in complex with ubiquitin (1XD3) with our cocrystal structure using US-align[45]. The normalized TM-score of these two structures is 0.97 and structural superimposition was shown in

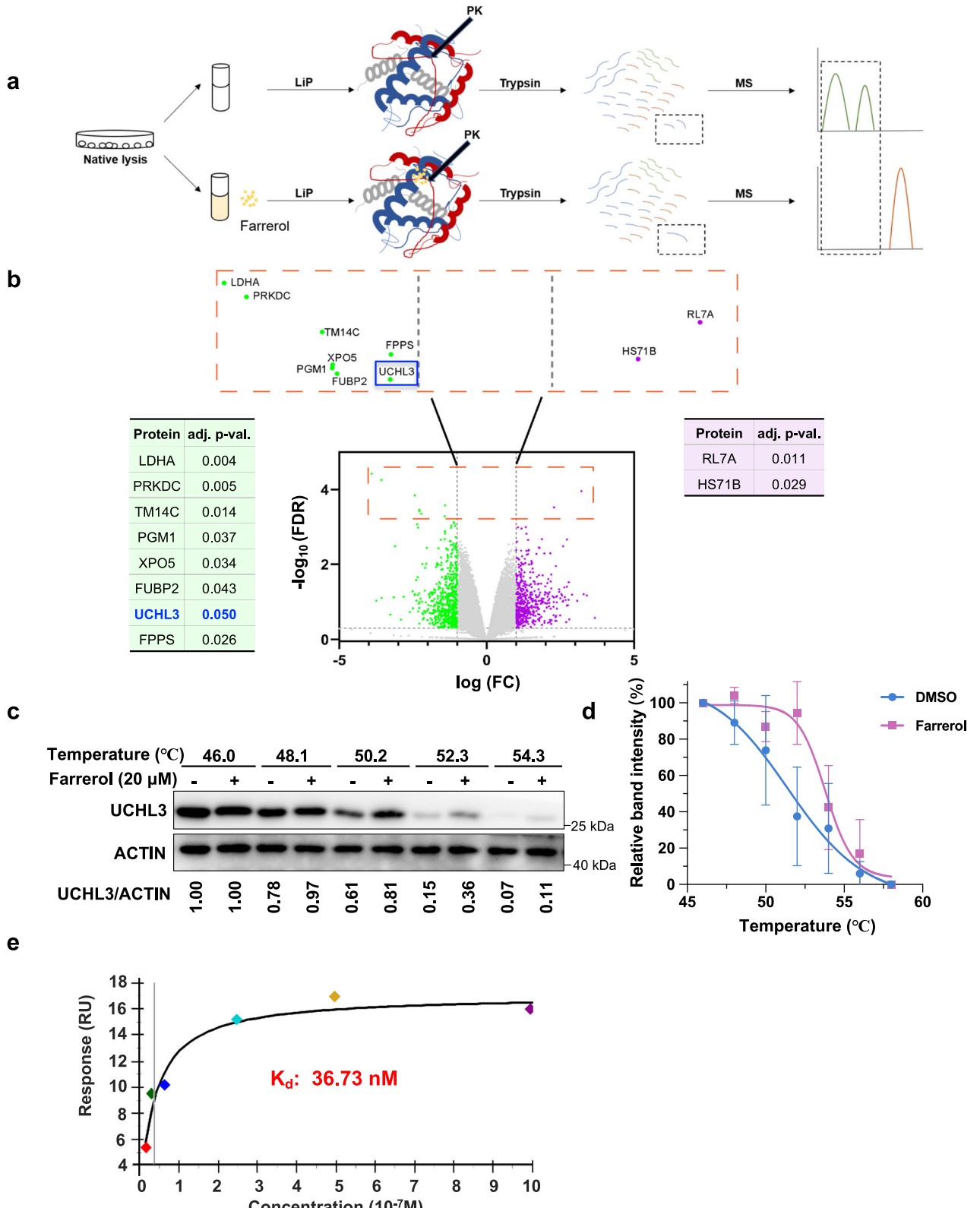

Supplementary Fig. 1e. The alignment indicated that farrerol did not affect the dynamic structure of UCHL3.

Then, we attempted to confirm whether the two amino acid residues were essential for the interaction between UCHL3 and farrerol through the use of CETSAs. We created two UCHL3 mutants, UCHL3–K187A and UCHL3–R215A, and then incubated farrerol with HEK293 cells that contained exogenously-introduced vectors in which

UCHL3–K187A or UCHL3–R215A was expressed. In contrast to the UCHL3 WT, an improvement in thermal stability was not observed for the two mutants (Fig. 2d–g), indicating that farrerol directly binds to UCHL3 through the two amino acid residues. Further SPR assay with the purified recombinant UCHL3–K187A R215A mutant protein and farrerol confirmed that mutating the two amino acid sites abolished the binding of farrerol to UCHL3 (Supplementary Fig. 1f).

**Fig. 1 | Systematic identification of UCHL3–farrerol interactions in native proteome extracts. a** Workflow of the LiP-SMap approach. The whole-cell lysates extracted under native lysis conditions were treated or not treated with farrerol. When farrerol binds to a protein, the local proteolytic susceptibility is altered. A limited proteolysis step is performed with proteinase K (PK) under native conditions, followed by complete digestion with trypsin under denaturing conditions to generate MS-measurable peptides. **b** Volcano plots of the LiP-SMap results. Each peptide is represented with a single data point in the graph. Proteins for which these peptides are mapped are listed in the order of fold change (FC) with its adjusted *P* value (adj. *P* val.). Statistical significance was determined by empirical Bayes *t* test, and was corrected for multiple comparisons by original FDR method of Benjamin and Hochberg. **c** Representative immunoblots for the CETSA assay carried out in the HEK293 cells treated with 20 μM farrerol (+) or DMSO (−). Western blots were developed using an anti-UCHL3 antibody. **d** CETSA curves of UCHL3 in the HEK293 cells were determined in the absence and presence of farrerol. Each band intensity of UCHL3 was normalized with respect to that obtained at 46 °C. The graphs are representative of three independent experiments. **e** SPR (BIAcore) measurements for the binding of farrerol to UCHL3. Data are presented as mean values ± s.e.m. for *n* = 3 (**d**). Source data are provided as a Source Data file.

## Farrerol activates the enzymatic activity of UCHL3 to promote HR repair

Since both farrerol and UCHL3 participate in regulating HR by improving the recruitment of RAD51 to DNA damage sites[28,36] and because we demonstrated that farrerol binds to UCHL3 with a high affinity, we wondered whether farrerol stimulated HR in a UCHL3-dependent fashion. We first confirmed that overexpressing UCHL3 promoted HR using the HR reporter cell line (Supplementary Fig. 2a, b). Then, we analyzed the efficiency of HR in control or UCHL3-depleted HCA2-H15c cells in the presence or absence of farrerol treatment. We found that depleting UCHL3 abolished the stimulation of HR by farrerol (Fig. 3a and Supplementary Fig. 2c). In line with this finding, we confirmed that the farrerol-mediated enhancement of HR repair was largely dependent on UCHL3 in mouse embryonic stem cells (mESCs), and this was determined using the extrachromosomal assay for analyzing HR efficiency (Fig. 3b).

Moreover, by counting the foci, we also examined RAD51 recruitment and γH2AX foci clearance at DNA damage sites in control or UCHL3-knockout cells treated with or without farrerol upon ionization irradiation (IR) at a dose of 2 Gy. Consistent with our previous report[36], we confirmed that in control HCA2-hTERT cells, farrerol promoted the recruitment of RAD51 at DNA damage sites (Fig. 3c, d) and accelerated γH2AX clearance at 4 h post IR (Fig. 3e, f); and RAD51 foci were cleared faster in farrerol-treated cells than in non-treated cells (Fig. 3c, d). In contrast, in HCA2-hTERT cells with UCHL3 depletion, the recruitment of RAD51 at DNA damage sites was impaired, and farrerol failed to stimulate recruitment at 4 h post IR damage; however, at 16 h post IR damage, RAD51 remained at the DNA damage sites, and farrerol lost its capacity to accelerate the clearance of RAD51 foci (Supplementary Fig. 2d). In addition, we obtained similar results that in a UCHL3-dependent manner farrerol promoted the recruitment of RAD51 and accelerated the clearance of γH2AX foci in cells irradiated by X-ray at a dose of 4 Gy (Supplementary Fig. 2e–h). Further immunostaining experiments revealed that farrerol stimulated the recruitment of RAD51 in IR-treated UCHL3-KO cells with UCHL3 WT restored but not in IR-treated UCHL3-KO cells with UCHL3–R215A/K187A mutant expressed (Fig. 3g, h). Taken together, these data indicated that farrerol promoted DNA repair by HR in a UCHL3-dependent manner.

Since the cocrystal structure of the UCHL3–farrerol complex revealed that farrerol directly binds to UCHL3 (Fig. 2a–c), a ubiquitin–AMC assay was then performed to determine whether farrerol activates UCHL3 enzymatic activity, and we found that farrerol affected both $K_m$ and $K_{cat}$, by slightly increasing $K_m$ from 47.72 to 56.27 nM and significantly increasing $K_{cat}$ from 11.99 to 19.64 s$^{-1}$ (Fig. 3i), suggesting that farrerol activates UCHL3 deubiquitinase activity. Moreover, we also performed a Ub-CHOP2 assay to test whether farrerol directly regulates the enzymatic activity of UCHL3. The data indicated that farrerol at a concentration of 0.1 μM significantly increased the deubiquitinase activity of UCHL3 (Supplementary Fig. 3a). Altogether, these in vitro results suggested that farrerol is an activator of UCHL3.

Since a previous report indicated that UCHL3 deubiquitinates RAD51 to promote its interaction with BRCA2, thereby simulating its recruitment at DNA damage sites and enhancing HR repair[28], we examined whether farrerol promoted the deubiquitination of RAD51 in a UCHL3-dependent manner. The in vitro deubiquitination assay revealed that farrerol promoted the UCHL3-mediated deubiquitination of RAD51 in a dose-dependent manner (Fig. 3j). Furthermore, farrerol treatment enhanced the deubiquitination of RAD51 in control cells, and knocking out UCHL3 abolished the farrerol-mediated stimulation in RAD51 deubiquitination (Fig. 3k), suggesting that farrerol promotes RAD51 deubiquitination through targeting UCHL3. We also performed co-IP experiments to examine the changes in the interaction between RAD51 and UCHL3 in HEK293 cells that were treated with farrerol, and we excluded the possibility that farrerol promoted the deubiquitination of RAD51 by enhancing the UCHL3-RAD51 interaction (Supplementary Fig. 3b, c).

Taken together, our data demonstrated that farrerol activated the deubiquitinase activity of UCHL3 to promote the deubiquitination of RAD51, thereby enhancing HR repair.

## Farrerol helps SCNT embryos overcome genetic barriers

Previous studies indicated that genetic abnormalities, such as DNA damage, a high frequency of DSBs and impaired chromosome integrity, constituted a barrier that greatly blocked the cell fate transition[2], in addition to well-known epigenetic barriers[46]. Moreover, these genetic barriers might not only cause early embryo development arrest but also severely inhibited their postimplantation development potentials[14]. Thus, we wondered whether farrerol-mediated HR enhancement could play a role in this process.

Here, we observed a higher proportion of chromosome copy number abnormalities in blastomeres in SCNT embryos from four-cell stage when compared with that of the IVF embryos (Fig. 4a, b and Supplementary Fig. 4a). In addition, we also detected the increase in genomic instability (Fig. 4c), measured by the alkaline comet assay, and the accumulation of DSBs (Fig. 4d), assayed by the immunostaining experiments with an anti-γH2AX antibody, in SCNT embryos at two-cell stage. Consequently, aneuploid karyotypes were frequently observed in morula SCNT embryos and SCNT-derived embryonic stem cells (SCNT-mESCs) (Fig. 4a, e and Supplementary Fig. 4b, c). Moreover, the number of γH2AX foci per embryo was significantly increased in the late developmental stage of SCNT embryos accompanied by the appearance of aneuploidy (Supplementary Fig. 4d). Collectively, these results suggest that genomic instability arises in nuclear transferred embryos at an early stage, and remains in high level throughout the reprogramming process.

We then hypothesized that the down-regulation in HR-directed repair might be the reason causing the rise in genomic instability in SCNT embryos. Indeed, we found that HR efficiency was significantly reduced in ESC lines of SCNT group than in those of the IVF group (Supplementary Fig. 4e). Consistently, the number of RAD51 foci in the nucleus was significantly reduced in the SCNT group at the two-cell stage than in the IVF group (Fig. 4f and Supplementary Fig. 4f). The transient treatment with mirin (-16 h), an inhibitor of MRE11 that is essential for the initiation of HR repair[47], significantly blocked SCNT-mediated reprogramming and retarded early embryonic development (Fig. 4g, h and Supplementary Table 1). Thus, these findings suggest

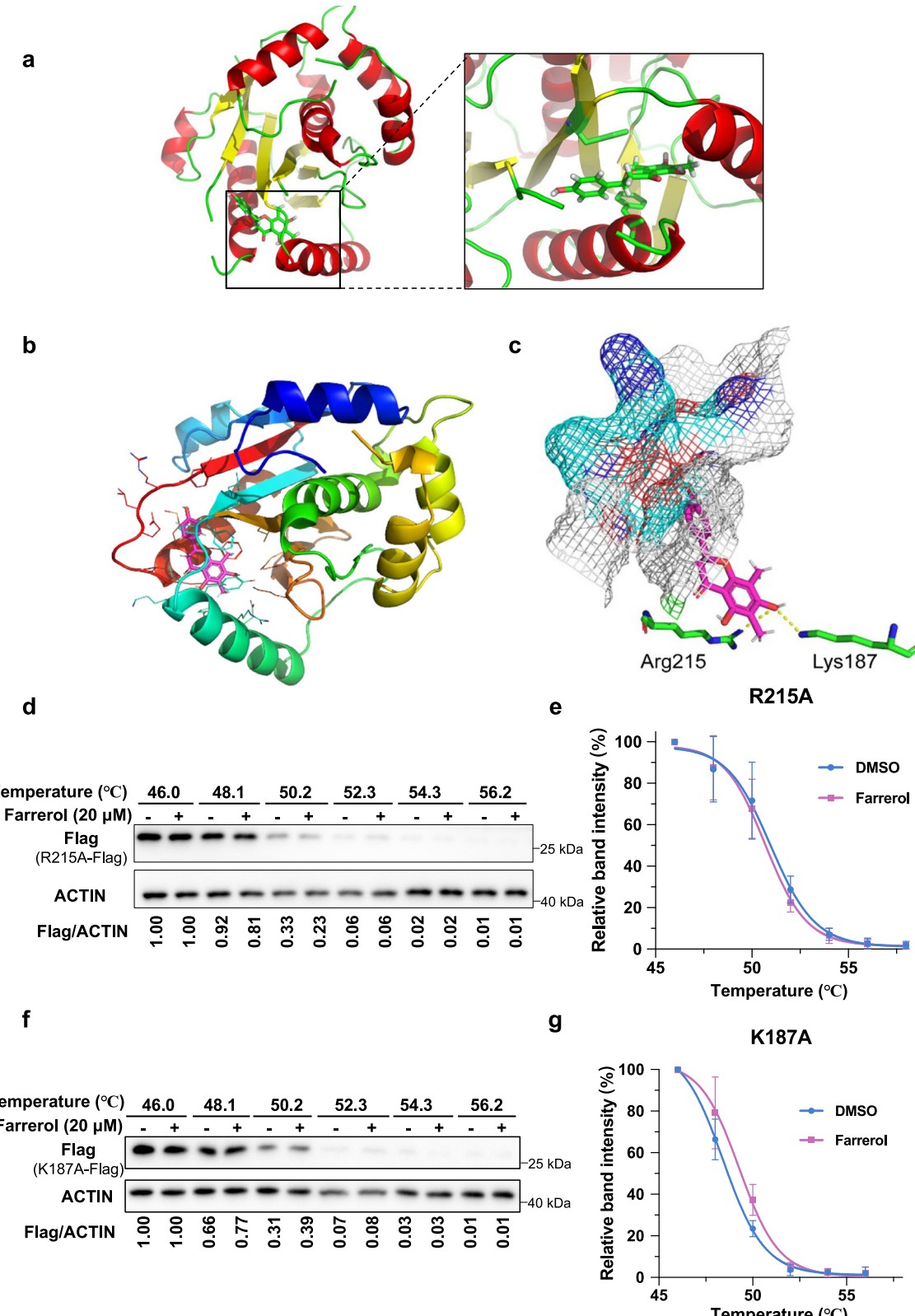

**Fig. 2 | Characterization and cocrystal structure of farrerol complexed with UCHL3. a** The structure of the UCHL3–farrerol complex. UCHL3 is shown as carbon atoms (red and yellow), and farrerol is depicted by sticks (green). **b** The crystal structure of UCHL3–farrerol is shown with a cartoon representation. The farrerol atoms are colored magenta and red. **c** Farrerol occupies the specific sites of UCHL3. The farrerol atoms are colored magenta and red. The pocket of UCHL3 is shown as a mesh. The hydrogen bonds are colored yellow. **d**, **e** Western blots and thermal denaturation curves of the main farrerol target UCHL3-R215 showing thermal stabilization upon farrerol treatment. **f**, **g** Western blots and thermal denaturation curves of the main farrerol target UCHL3-K187 showing the thermal stabilization that occurred upon farrerol treatment. Data are presented as mean values ± s.e.m. for $n$ = 3 (**e**, **g**). Source data are provided as a Source Data file.

**Table 1 | Data collection and refinement statistics (molecular replacement)**

| | UCHL3–Farrerol UCHL3–Farrerol |
|---|---|
| **Data collection** | |
| Space group | $P2_12_12_1$ |
| Cell dimensions | |
| $a, b, c$ (Å) | 47.054, 63.87, 77.16 |
| $\alpha, \beta, \gamma$ (°) | 90, 90, 90 |
| Resolution (Å) | 21.8–1.58 (1.63–1.58) |
| $R_{sym}$ or $R_{merge}$ | 0.158 (0.833) |
| $I / \sigma I$ | 20.67 (1.25) |
| Completeness (%) | 96.76 (90.63) |
| Redundancy | 12.5 (9.3) |
| **Refinement** | |
| Resolution (Å) | 1.58 |
| No. of reflections | 31512 |
| $R_{work}/R_{free}$ | 17.9/18.04 |
| No. of atoms | |
| Protein | 1623 |
| Ligand/ion | 38 |
| Water | 261 |
| $B$-factors | 27.15 |
| Protein | 25.88 |
| Ligand/ion | 21.46 |
| Water | 35.56 |
| R.m.s. deviations | |
| Bond lengths (Å) | 0.014 |

that effective HR is essential to early embryonic development, but it is inefficient in SCNT embryos.

Given that farrerol can significantly improve the efficiency of HR repair, as well as the HR pathway relying mainly on sister chromatids present in the S and G2 phases as repair templates[48], we then opted to perform a farrerol treatment when these manipulated SCNT embryos began to enter the S phase of the first cell cycle post-$Sr^{2+}$ activation (Fig. 4a). Strikingly, a short-term farrerol treatment (-16 h) significantly enhanced genomic stability (Fig. 4c, d), HR efficiency (Supplementary Fig. 4e), RAD51 foci number in the nucleus (Fig. 4f and Supplementary Fig. 4f), and euploidy maintenance in both the SCNT embryos and SCNT-derived ESCs (Fig. 4e, Supplementary Fig. 4b, c). More importantly, we demonstrated that the farrerol treatment markedly increased the preimplantation development potential of SCNT embryos by approximately twofold (Fig. 4g, h). A concentration of 0.05 μM was then set as the optimal concentration after precisely testing the concentration gradients (Supplementary Fig. 4g, h). Notably, almost 90% of two-cell embryos could normally develop to the four-cell stage, greatly improving the well-known two-cell block, an inherent transition barrier during somatic reprogramming in SCNT embryos.

In conclusion, HR repair is inefficient in SCNT embryos, and farrerol, as an effective regulator of HR, could promote SCNT embryos to overcome the genetic barrier and maintain genome stability, thus enhancing reprogramming efficiency.

### Farrerol ameliorates the insufficient transcriptional and epigenetic reprogramming in SCNT embryos

To uncover how the activation of HR repair by farrerol influences the transcription network, we collected SCNT embryos with or without a farrerol treatment for RNA-seq analysis; notably, wild-type IVF embryos were used as the control (Fig. 4a). The results clearly showed

that the dysregulated reprogramming resistant regions (RRRs) in SCNT was partially restored after farrerol treatment (Fig. 5a). Meanwhile, an upregulation in the representative zygotic genome activation (ZGA) related genes such as *Dux* and *Zscan4* cluster was observed (Fig. 5b). The recovery at the ZGA stage might further enhance an effective transcriptional reprogramming thereafter, and promote later embryonic development, which partly explains the increased blastocyst rate by farrerol treatment (Fig. 4g, h). Accordingly, when comparing the transcriptomes of each group at the morula stage, we observed that the farrerol treatment changed the gross gene expression pattern in the SCNT embryos to a similar pattern as that in the IVF group (Supplementary Fig. 5a). The genes that are important for embryonic development and further differentiation were upregulated (Cluster I), whereas the aberrant somatic identity (such as *Id1*, inhibitor of DNA binding 1 and *Sox4*, SRY-Box transcription Factor 4) was largely erased (Cluster II) in farrerol-treated group (Supplementary Fig. 5b, c).

Specifically, embryonic polarity, a feature that contributes to the normal progression of lineage differentiation, was significantly improved in the farrerol-treated group, as demonstrated by the high expression of polarization-related genes, such as *Klf5* and *Arpc1* (Fig. 5c). In addition, *Nanog* and *Cdx2*, which are important marker genes of inner cell mass (ICM) and trophectoderm (TE) lineages, respectively, showed a much higher expression pattern in the farrerol-treated group than in the non-treated group (Fig. 5d–f). Besides, more ICM numbers and recovered TE/ICM ratios were observed in the farrerol-treated SCNT blastocysts, indicating that the first lineage differentiation potential in the SCNT embryos is superior after the addition of farrerol (Supplementary Fig. 5d, e). Consistent with the restoration of lineage genes, target gene expression of NANOG and CDX2 was also effectively improved in the farrerol-treated group (Supplementary Fig. 5f, g). These data demonstrate that farrerol treatment at the early stage could improve the accuracy of the developmental trajectory in the late SCNT embryonic stage.

In addition to the pluripotent transcriptome recovery, more strikingly, we found that the farrerol treatment could further promote epigenetic reprogramming during SCNT process. A higher enrichment of the histone variant H3.3 was observed in IVF and SCNT embryos that were treated with farrerol (Fig. 5g and Supplementary Fig. 6a), suggesting that efficient HR repair might help remodel chromatin, which was also confirmed by the repair in RRRs. However, this kind of chromatin opening did not induce a significant global 5-Methylcytosine (5-mC) change (Supplementary Fig. 6b). More interestingly, the repressive histone modification H3K27me3, which is closely related to the regulation of gene expression during lineage specification, was significantly enriched (from 7.14 to 50%) in the ICM of farrerol-treated SCNT embryos (Fig. 5h and Supplementary Fig. 6c; Supplementary Table 2). It was reported that high levels of H3K27me3 at bivalent genes play a key role in preventing the premature activation of bivalent genes, which further guarantees that the postimplantation embryo undergoes lineage differentiation[49]. Accordingly, farrerol-treated SCNT embryos further successfully differentiated into *Gata6*-positive primitive endoderm (PrE) cells, whereas this ratio was significantly reduced in the non-treated SCNT group (Fig. 5i and Supplementary Fig. 6d, e). In addition to *Nanog*, *Cdx2* and *Gata6*, we also observed that the expression of other lineage differentiation-related genes, including *Klf4*, *Esrrb*, *Gata3*, and *Pou5f1*, was significantly improved in the farrerol-treated group (Supplementary Fig. 6f).

Collectively, farrerol ameliorated the impaired transcriptional and epigenetic reprogramming of SCNT embryos and contributed to a proper lineage specification.

### Farrerol improves SCNT efficiency, which is largely dependent on UCHL3

To further test the effect of farrerol on the postimplantation ability of SCNT embryos, we transplanted the farrerol-treated

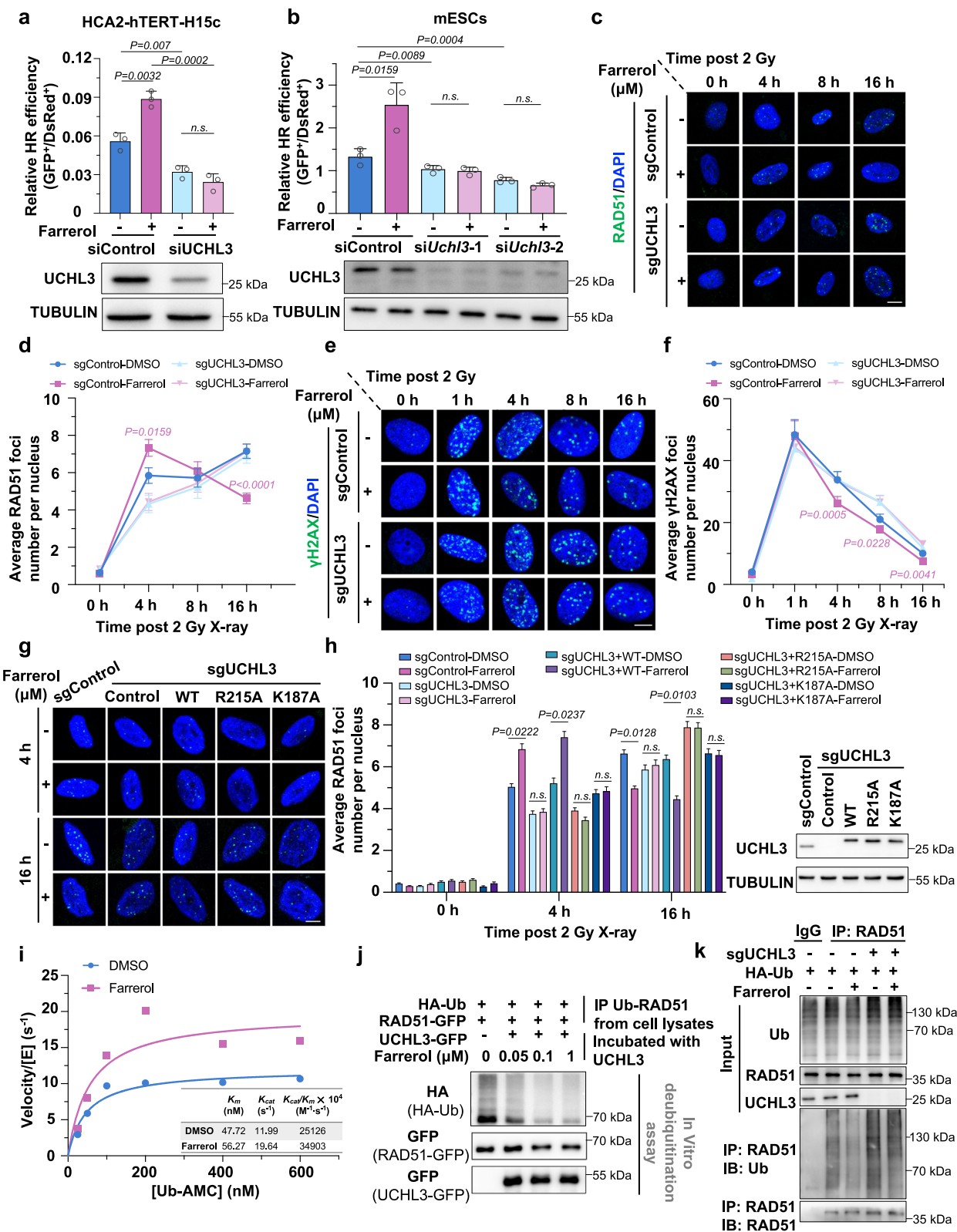

SCNT embryos into pseudopregnant recipient females at the two-cell stage (Fig. 4a). Although only 0.2% of the transferred non-treated SCNT 2-cells developed into full-term E19.5 pups, this proportion was significantly increased by approximately 10–20-fold after farrerol treatment (Supplementary Table 3). In addition, the average body weight of the newborn pups derived from the

farrerol-treated SCNT embryos was indistinguishable from that of IVF-derived pups, and the pups could undergo healthy maturation into fertile adults (Supplementary Fig. 7a, b). More strikingly, farrerol treatment effectively recovered the persistent hyperplasia of SCNT placentae[16,17]. In particular, the weight of placentae produced by SCNT embryos was partially reduced by farrerol

**Fig. 3 | Farrerol-mediated activation of HR is dependent upon UCHL3 enzyme activity. a** Effects of farrerol treatment and UCHL3 depletion on HR. **b** Effects of UCHL3 depletion on HR followed by DMSO or farrerol treatment in mouse ESCs. Western blot analysis of UCHL3 in the control and UCHL3-knockdown cells was performed. **c** Representative images of RAD51 recruitment in the UCHL3-knockout HCA2-hTERT cells after the 2 Gy X-ray treatment. Scale bar: 5 μm. **d** The recruitment of RAD51 in UCHL3-knockout HCA2-hTERT cells with farrerol treatment. At least 40 cells were counted for each group. **e** Representative images of γH2AX foci in the control and UCHL3-knockout cells at indicated time points post IR (2 Gy). Scale bar: 5 μm. **f** Quantification of positive cells with γH2AX foci in the control and UCHL3-knockout cells at indicated time points post IR (2 Gy), and at least 40 cells were counted for each group. **g**, **h** UHCL3 knockout cells transfected with vectors encoding UCHL3 WT, UCHL3–R215A or UCHL3–K187A, were irradiated at a dosage of 2 Gy, and the average number of RAD51 foci at indicated time points was

quantified. The expression of UCHL3 WT or UCHL3 mutants (R215A or K187A) in UCHL3 knockout cells is shown in the right panel. At least 40 cells were counted for each group. **i** Progress curve for the UCHL3-catalyzed hydrolysis of UB-AMC in the presence of farrerol at a concentration of 0.1 μM. The solid line was drawn using the Michaelis–Menten equation and the following best-fit parameters: $K_m = 4.772 \times 10^{-8}$ M (DMSO) and $K_m = 5.627 \times 10^{-8}$ M (farrerol). **j** In vitro deubiquitination analysis of UCHL3 in the presence or absence of farrerol. **k** Examination of RAD51 ubiquitination in UCHL3-knockout cells treated with farrerol. Data are presented as mean values ± s.d. for $n = 3$ (**a**, **b**). Data are presented as mean values ± s.e.m. (**d**, **f**, **h**). Statistical significance was calculated with a two-tailed unpaired $t$ test (**a**, **b**, **d**, **f**, **h**). n.s. not significant. Experiments were repeated three times independently; data of one representative experiment are shown (**j**, **k**). Source data are provided as a Source Data file.

treatment[50] (Supplementary Fig. 7c). Moreover, the structure of these placentae from farrerol-treated SCNT embryos was more organized, demonstrating three-layered smooth edges, which provided structural integrity (Supplementary Fig. 7d). Notably, the restored expression of noncanonical (such as *Slc38a4* and *Sfmbt2*) and canonical (such as *H19* and *Igf2*) imprinted genes could be detected in extraembryonic ectoderm in farrerol-treated group (Supplementary Fig. 7e, f). Therefore, the placental hyperplasia frequently observed in SCNT embryos could be partially improved by farrerol, and this improvement manifested a reduction in weight, a compact structure, and an efficient compression of the spongiotrophoblast (ST) layer, in which the effect by rescued expression of canonical and noncanonical imprints might be significant.

To verify whether farrerol improved SCNT efficiency by targeting UCHL3, we depleted UCHL3 by injecting both sgRNA and siRNA against the *Uchl3* gene into enucleated oocytes before nuclear transfer (Fig. 6a and Supplementary Fig. 8a). The result showed that *Uchl3* depletion did greatly abrogate the farrerol-induced increase in the development rates of SCNT embryos (Fig. 6b, c and Supplementary Fig. 8b), whereas it did not affect the development of IVF embryos (Supplementary Fig. 8c). Specifically, the four-cell and blastocyst development rates after *Uchl3* depletion were only 70% and 52.78%, respectively. As a comparison, this ratio was as high as 90.08% and 69.44% in wild-type farrerol-treated SCNT embryos, which indicated that UCHL3 was an essential target of farrerol and led to improved SCNT efficiency (Fig. 6b, c and Supplementary Table 1). Moreover, farrerol treatment reduced the amounts of DSBs, as indicated by γH2AX foci in two-cell stage of control SCNT embryos, while *Uchl3* depletion abolished the farrerol-mediated inhibition of the amounts of DSBs (Fig. 6d). Besides, the UCHL3-deficient SCNT blastocysts did not exhibit increased ICM and primitive endoderm (PrE) cell numbers or recovered *Gata6* expression after farrerol treatment (Fig. 6e–g). More strikingly, the enrichment of H3K27me3 in farrerol-treated SCNT blastocysts was also impaired, with the percentage of H3K27me3-positive ICM decreased from 43.75 to 20% (Fig. 6h and Supplementary Table 2). Next, to test whether *Uchl3* depletion further affected the postimplantation development potential of farrerol-treated SCNT embryos, we transplanted these embryos into pseudopregnant mice. Strikingly, the depletion of *Uchl3* completely diminished the positive effect of farrerol on producing cloned mice, as no 173 UCHL3-deficient farrerol-treated two-cell SCNT embryos could normally develop into a term E19.5 stage (Supplementary Table 3).

Taken together, these results confirmed that improving the development of SCNT embryos by employing farrerol was largely dependent on UCHL3, the protein that farrerol targets.

## Discussion

In this report, we demonstrated that farrerol is a direct and potent activator of UCHL3. The cocrystal structure and SPR results reveals

that farrerol binds to two amino acid residues, R215 and K187, of UCHL3. We performed in vitro and in vivo deubiquitination assay to establish that farrerol directly stimulates UCHL3 enzyme activity and promotes HR by UCHL3-RAD51 axis. In addition, HR is essential for the maintenance of genomic integrity[51], whereas impaired HR leads to aneuploidy and cell cycle arrest[21]. We demonstrated that SCNT embryos showed severely impaired HR repair, genomic instability and aneuploidy. Farrerol could activate HR through UCHL3 and further modify transcriptional and epigenetic network, thus improving the efficiency of SCNT during preimplantation and postimplantation development.

Previous studies have revealed that farrerol exhibits various biological activities, including antibacterial, anti-inflammatory, antioxidant and pro-vasoactive effects[37–41], but how farrerol regulates these biological processes remains to be further determined. In the report, the LiP-SMap assay indicated that in addition to UCHL3, farrerol might potentially target a number of other proteins when performing its pharmacological role. For instance, farrerol might target LDHA to participate in the regulation of oxidative metabolism, thereby exhibiting its antioxidant activity. Farrerol might also regulate tumor suppression by targeting XPO5. Further thorough studies on whether farrerol directly targets these proteins and the underlying mechanisms might open up new avenues for the application of farrerol when treating other types of diseases.

Notably, several reports also indicated that UCHL3 is aberrantly overexpressed in different types of tumors, so caution should be taken when utilizing farrerol, as the farrerol-mediated activation of UCHL3 might be dangerous for cancer patients. However, targeting UCHL3 to suppress HR repair, which is also often abnormally upregulated in cancer cells, might be a novel approach to treat cancer. Indeed, in previous work, it was reported that a UHCL3 inhibitor was identified and could be further applied in treating non-small lung cancer, breast cancer and lung cancer;[35,52] however, whether the suppressive effect of these inhibitors on UHCL3 occurs through the direct binding and targeting of UCHL3 remains to be determined. Since we demonstrated that farrerol binds to UCHL3 through two amino acid residues to activate its enzymatic activity, in order to treat HR-overactivated cancer or overcome radio/chemo-resistance, it is likely worthwhile to try generating farrerol derivatives that exhibit an inhibitory effect on UCHL3 while still retaining a high affinity to UCHL3.

It was reported that genomic stability is critical in the dynamic process of somatic cells reprogramming into iPSCs[53] and is essential for the pluripotency of ESCs and iPSCs, whereas chromosomal aneuploidy results in a sharp decrease in cell differentiation potentials in vivo[54,55]. By applying SCNT as a model, we revealed the importance of genomic stability in the successful acquisition of totipotency. Here, we demonstrated that the enhanced HR efficiency of farrerol was primarily dependent on UCHL3 deubiquitinating RAD51. However, the effect of farrerol is probably not limited to the activation of HR pathway. It could also partially promote the open chromatin state, the

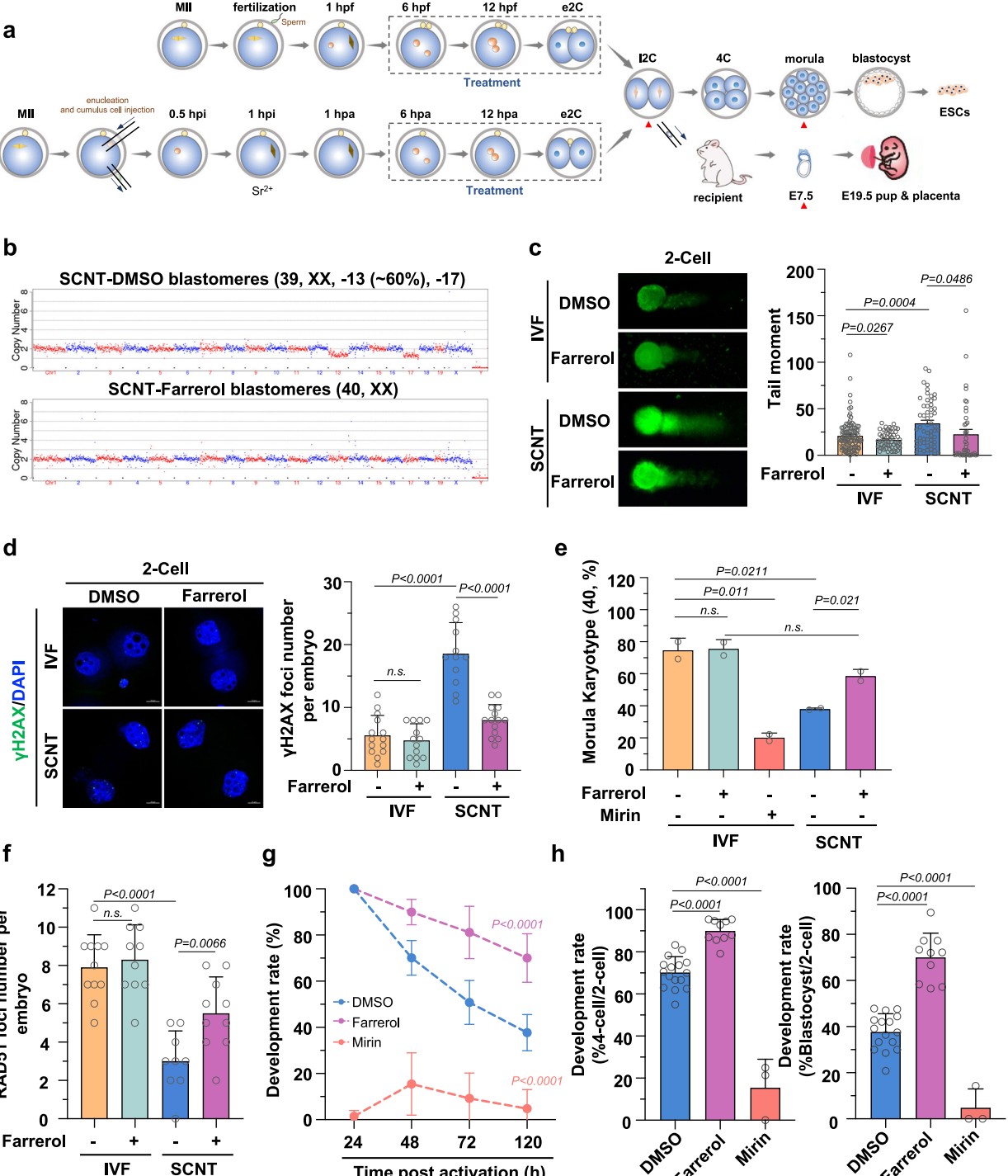

**Fig. 4 | Farrerol improved SCNT efficiency. a** Schematic diagram showing the preparation of mouse IVF and SCNT embryos treated with DMSO or farrerol for transplantation, in vitro development analysis, RNA-seq (marked by red triangle) and derivation of ESC lines. e2C: early two-cell stage, l2C: late two-cell stage, 4C: four-cell stage, hpf: hours post fertilization, hpi: hours post injection, hpa: hours post activation, E7.5/19.5: embryonic day 7.5/19.5. **b** Representative images of copy number variation (CNV) analysis in SCNT embryos by indicated treatment. SCNT embryos without farrerol showed higher level CNV abnormality. In total, 1–2 blastomeres from embryos at the four-cell stage were randomly picked for the examination. **c** Representative photographs (left) and the average tail moment by comet assay (right) detecting two-cell stage embryos in IVF or SCNT group treated with DMSO or 0.05 μM farrerol. At least 40 cells were counted for each group. **d** Immunofluorescence of two-cell embryos stained with anti-γH2AX antibodies (green) and DAPI (blue). The average γH2AX foci number per embryo is shown in

the right (*n* = 12–14 embryos). Scale bar: 10 μm. **e** The average euploidy rate of morula cells in the IVF group treated with DMSO, 0.05 μM farrerol or 50 μM mirin and SCNT group treated with DMSO or 0.05 μM farrerol (*n* = 2 independent experiments, at least eight embryos were analyzed for each experiment). **f** The plot showing the number of RAD51 foci in two-cell embryos from IVF or SCNT group treated with DMSO or 0.05 μM farrerol (*n* = 9–11 embryos). **g** Development rate of SCNT embryos treated with indicated small molecules (*n* = 3–16 embryos). The concentration of farrerol and mirin was 0.05 μM and 50 μM, respectively. **h** Development rate of four-cell embryos and hatching blastocysts at E4.5 treated with indicated small chemicals during SCNT process (*n* = 3–16 embryos). Data are presented as mean values ± s.e.m. (**c**). Data are presented as mean values ± s.d. (**d**–**h**). Statistical significance was calculated with a two-tailed unpaired *t* test (**c**–**h**). n.s. not significant. Source data are provided as a Source Data file.

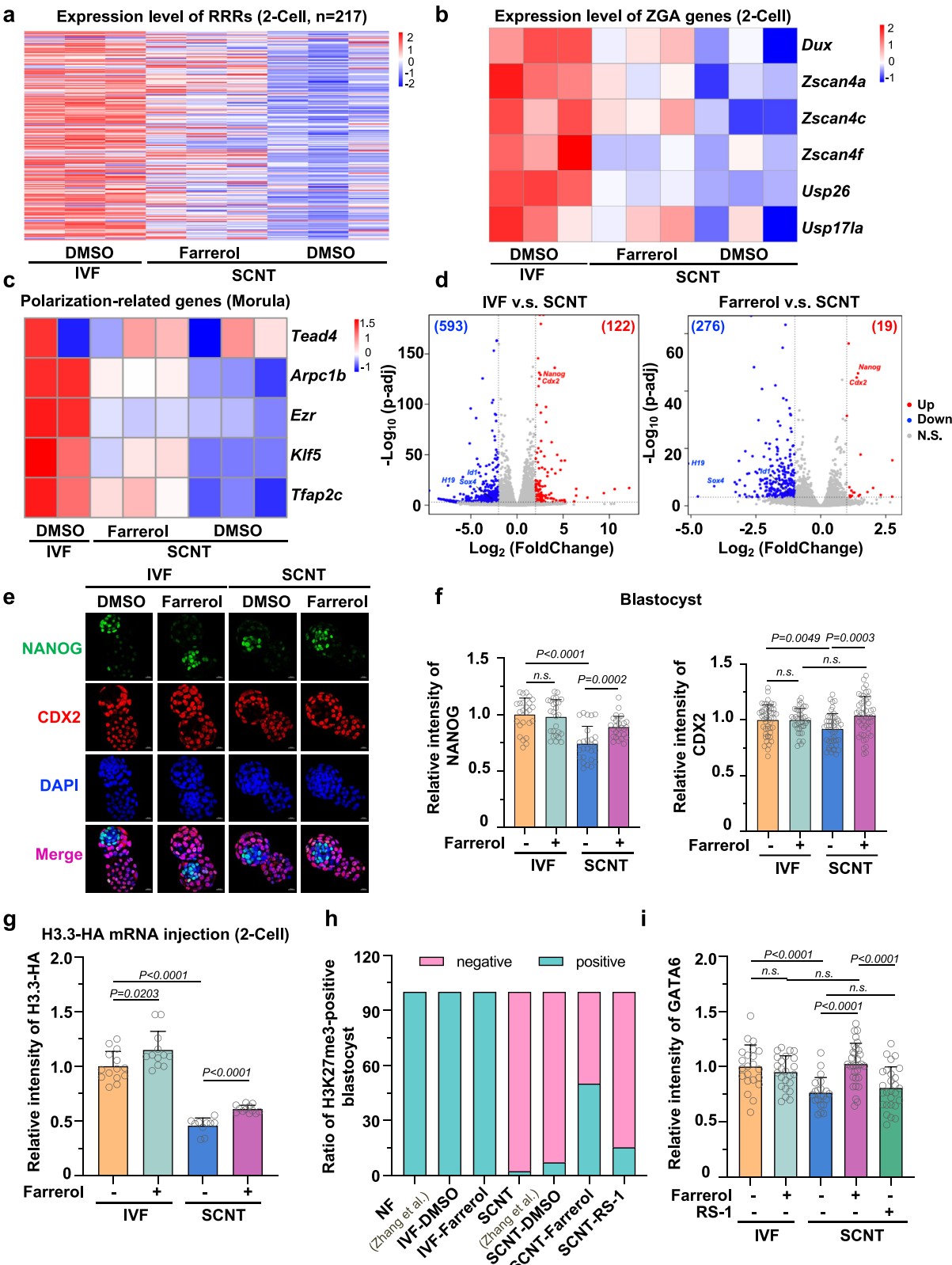

**a** Expression level of RRRs (2-Cell, n=217)

**b** Expression level of ZGA genes (2-Cell)

**c** Polarization-related genes (Morula)

**d** IVF v.s. SCNT    Farrerol v.s. SCNT

**e**

**f** Blastocyst

**g** H3.3-HA mRNA injection (2-Cell)

**h**

**i**

activation of RRRs, the establishment of H3K27me3 modification, and the expression of lineage differentiation genes. The additional functions of farrerol might help explain why directly activating RAD51 through RS-1 appeared to have a limited effect on SCNT-mediated reprogramming.

In several studies, it was suggested that EZH2-mediated H3K27me3 may function as an important mechanism to regulate transcription both at lineage-specific genes and DNA damage sites[56,57]. In addition to H3K27me3, other epigenetic modifications have also been reported to be closely related to repairing DNA damage. For example, the demethylation of H3K4me3 is an important step in promoting the recruitment of ZMYND8-NuRD chromatin remodeling complexes at DSB sites, which have been shown to locally repress transcription around DSB sites and promote HR[58]. Moreover, TET3-

**Fig. 5 | Farrerol improves SCNT embryo transcriptome and epigenetic barriers.**
**a** Heatmap illustrating the transcription levels of 217 reprogramming resistant regions (RRRs) in two-cell stage embryos from IVF groups and SCNT group with or without farrerol treatment. The sites of RRRs are cited from Matoba's paper[14]. **b** Heatmap showing the expression comparison of representative ZGA related genes in two-cell stage from IVF embryos and SCNT embryos with or without farrerol treatment. **c** Heatmap showing the expression of polarization-related genes in indicated groups. **d** Volcano plots showing the differentially expressed genes (DEGs) between IVF and non-treated SCNT groups (left), and between farrerol-treated and non-treated SCNT groups (right). DEGs between IVF and SCNT: adjusted $P$ value <0.001, fold change >4; DEGs between farrerol-treated SCNT and SCNT: adjusted $P$ value <0.001, fold change >2. Selected genes are labeled with gene symbols. **e** Immunofluorescent staining of NANOG (green), CDX2 (red) and

DAPI (blue) in indicated blastocysts. Scale bar: 20 μm. **f** Plot showing the comparison of relative fluorescence intensity of NANOG (left) ($n = 25$–27 blastocysts), and CDX2 (right) ($n = 41$–49 blastocysts) among indicated groups, as related to (**e**). **g** Plot showing the relative fluorescence intensity of H3.3-HA per nucleus in IVF and SCNT embryos at two-cell stage with or without farrerol treatment ($n = 10$–14 embryos). **h** Bar graph showing the ratio of H3K27me3-positive-ICM blastocysts in indicated groups. Data from naturally fertilization (NF) and SCNT embryos are cited from the published data as shown[69]. **i** Plot showing the relative fluorescence intensity of GATA6 in blastocysts among indicated groups ($n = 21$–31 blastocysts). Data are presented as mean values ± s.d. (**f, g, i**). Statistical significance was calculated with a two-tailed unpaired $t$ test (**d, f, g, i**). n.s. not significant. Source data are provided as a Source Data file.

mediated DNA demethylation can respond to DNA damage and promote ATR[59]. Taken together, epigenetic modifications not only regulate gene expression but also play important roles in maintaining genomic stability. On the other hand, aberrant DSB repair might cause relocalization of chromatin modifiers (the "RCM" hypothesis), alter the epigenome landscape and advance the epigenetic clock[60,61]. Here, we showed that farrerol have multiple contributions in improving SCNT efficiency, which is an interesting topic for future research. In addition, whether breaking epigenetic barriers also influences the DNA repair pathway or partially functions in maintaining genomic stability needs to be further investigated.

Since the method of stabilizing genomes by efficient DNA repair may not only be beneficial to improving SCNT efficiency but also contribute to a number of other biological processes, such as delaying the onset of aging and tumor prevention, it is worth testing the roles of farrerol as a genome stabilizer in these processes. In fact, previous pharmacokinetic studies on farrerol in both mice and rats revealed its potential as a therapeutic drug, although the safety and efficacy of farrerol must be further determined. In addition, although several lines of evidence have indicated that UCHL3 plays positive roles in preventing neurodegeneration[62], retinal degeneration[63], and substantial memory impairment[64], whether the activation of UCHL3 could contribute to the cure of these age-related diseases needs to be further tested. The generation of UCHL3-overexpressing mice and studies with these mice on aging or tumorigenesis might be necessary before further testing the roles of farrerol in alleviating the syndromes of these age-related diseases, delaying the onset of aging, and preventing tumorigenesis.

## Methods

### Limited proteolysis combined with mass spectrometry (LiP-SMap)
LiP-SMap was performed according to a previous report[42]. Briefly, cell lysates containing approximately 500 μg of total protein sample were incubated with farrerol or vehicle (DMSO) for 10 min at 25 °C. Proteinase K (5 μg, Sigma-Aldrich) was added simultaneously to each sample for the following 10 min. Then, the reaction was stopped by heating at 98 °C for 3 min. The samples were subjected to limited proteolysis to generate structure-specific protein fragments. Complete digestion was performed with trypsin (Promega) at a 1:50 ratio (trypsin:protein, w-w) for 16 h at 37 °C, and then the reaction was stopped by adding formic acid to reach a pH less than 2. The polypeptide samples were dissolved in 0.1% formic acid and detected by Orbitrap mass spectrometry with label-free quantitation (LFQ) analysis to search for differential proteins on the MaxQuant platform via MS sensitivity.

### Surface plasmon resonance (SPR) measurements
Real-time surface plasmon resonance[65] analyses were performed on a Biacore T200 apparatus (GE Healthcare) at 25 °C in PBS-P + buffer (0.02 M phosphate buffer, 2.7 mM KCl, 13.7 mM NaCl, 0.05% Surfactant P20 and 5% DMSO). UCHL3, which was diluted at 10 μg mL⁻¹ in

10 mM acetate at pH 4, was covalently immobilized at 10000 resonance units on a CM5 sensor chip (GE Healthcare) using an amine coupling protocol according to the manufacturer's instructions. Affinity experiments with farrerol were performed at a flow rate of 30 μL min⁻¹ with twofold serial dilutions (15.625–2000 nM). All curves were evaluated (Biacore T200 Evaluation Software 3.0; GE Healthcare) after double-referencing subtraction using a bivalent fitting model.

### Crystallization and structure determination
The final UCHL3 protein was purified with 20 mM Tris-HCl pH 7.5 and 100 mM NaCl and mixed with the compound at a 1:1.5 molar ratio. After incubation at 4 °C for 1 h, the complex of the compound and UCHL3 was crystallized using the sitting drop method. After one week, the best crystal was obtained under 1.4 M sodium citrate tribasic dihydrate and 0.1 M HEPES sodium at pH 7.5. The diffraction data were collected in the X-ray crystallography facility at Tsinghua University (XtaLAB Synergy Custom FRX and a hybrid photon counting detector HyPix-6000, Rigaku, Japan). The best crystal diffracted to 2.5 Å at 100 K. The data processing was performed using the Crysalis Pro program with the $P2_12_12_1$ space group.

The structures were solved by molecular replacement with Phenix software (v8.3) using a modified version of the structure 1UCH. Initial phases were improved by rigid body refinement followed by rounds of simulated annealing and anisotropic B-factor refinement using the Phenix suite. The structures were completed in alternative cycles of manual model building in COOT and restrained TLS refinement in Phenix.

### Cell culture and transfection
Human embryonic kidney epithelial cells (HEK293, ATCC, CRL-1573), HCA2-hTERT cells[66] (an immortalized foreskin fibroblast cell line) and HCA2-hTERT-H15c cells[66] were cultured in DMEM (Sigma, Cat. # D6429) supplemented with 10% (v/v) heat-inactivated fetal bovine serum (Gibco, Cat. #16000-044), 1% (v/v) penicillin–streptomycin (Gibco, Cat. #15140-122), and 1% MEM nonessential amino acids (Gibco, Cat. #11140-050). The cells were maintained at 37 °C in a 5% $CO_2$ atmosphere.

The control- and SCNT-mESCs were maintained in DMEM supplemented with 15% FBS, 1% nonessential amino acids, 1% penicillin/streptomycin, 1% nucleosides, 1% L-glutamine, 0.1 mM mercaptoethanol, 1000 U/mL LIF, 1 mM PD0325901 (Selleck), and 3 mM CHIR99021 (Selleck).

The HCA2-hTERT cells were seeded at $5 \times 10^5$ cells per 10-cm dish. After 24 h, the cells were pretreated with DMSO or the indicated concentration of farrerol for 24 h. The cells were then transfected with the indicated plasmids or siRNAs with the DT130 program in a Lonza 4D instrument. Forty-eight hours post transfection, the cells were harvested for FACS analysis on a BD FACSVerse instrument (BD Biosciences, San Jose, CA). The FACS data were analyzed by FlowJo7.6 (Ashland, OR, USA), and DNA repair efficiency was calculated as the

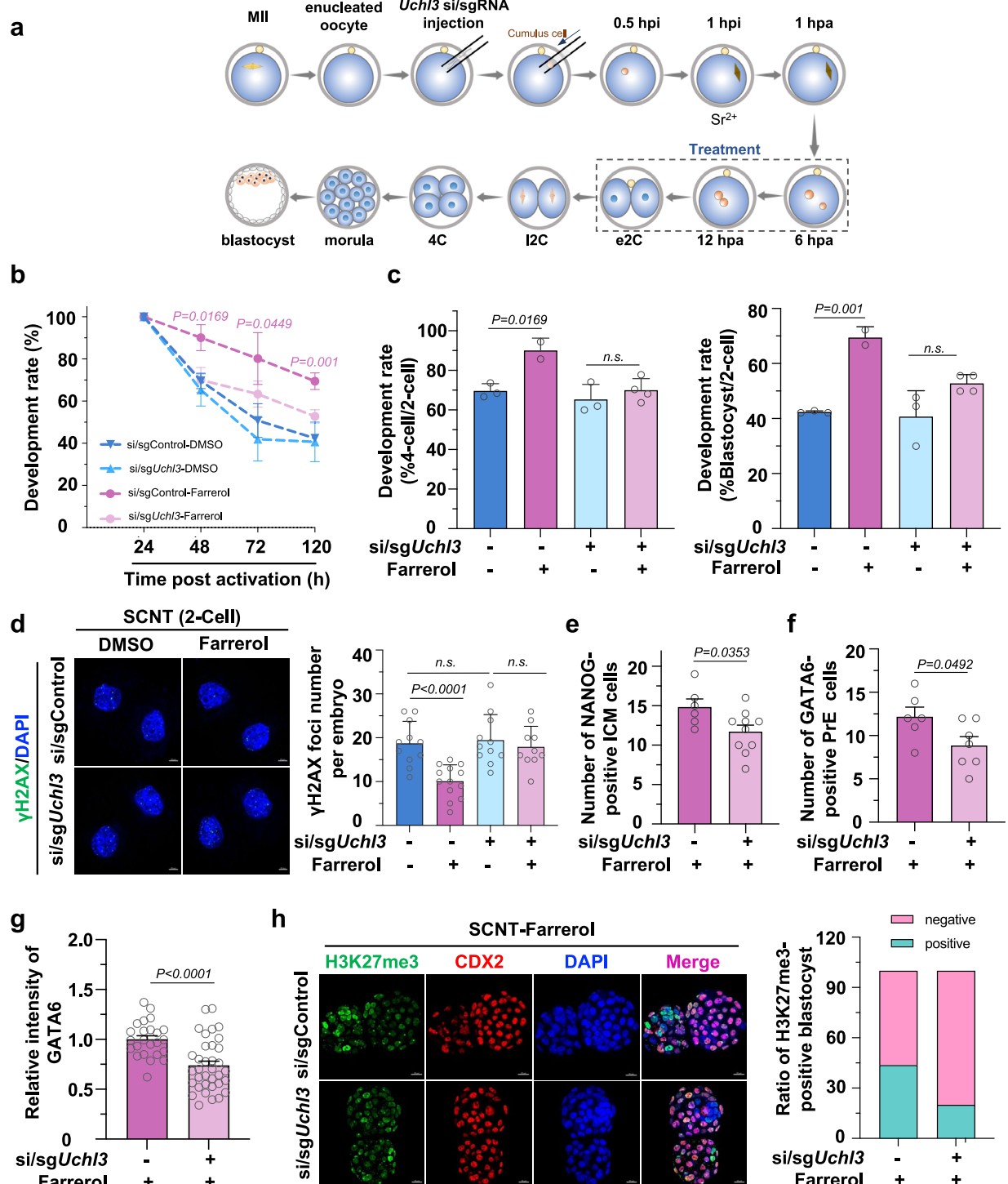

**Fig. 6 | Farrerol improved SCNT efficiency through UCHL3. a** Illustration of the experimental design for the injection of *Uchl3* siRNAs and sgRNAs in SCNT embryos following farrerol treatment. e2C: early two-cell stage, l2C: late two-cell stage, hpi: hours post injection, hpa: hours post activation. **b** Development rate of SCNT embryos treated with DMSO or farrerol following *Uchl3* depletion (*n* = 2–4 embryos). **c** Development rate of four-cell embryos (left) and hatching blastocysts at E4.5 (right) in indicated groups related to (**b**) (*n* = 2–4 embryos). **d** Representative images (left) and quantification (right) of γH2AX foci in *Uchl3*-depleted two-cell embryos with or without farrerol treatment (*n* = 10–13 embryos). Scale bars, 10 μm. **e** Plot showing the number of Nanog-positive ICM cells after farrerol treatment following *Uchl3* depletion (*n* = 6–10 cells). **f** Plot showing the number of GATA6-

positive primitive endoderm (PrE) cells after farrerol treatment following *Uchl3* depletion (*n* = 6–7 cells). **g** Plot showing the relative fluorescence intensity of GATA6 in blastocysts after farrerol treatment following *Uchl3* depletion (*n* = 24–35 blastocysts). **h** Representative images of immunofluorescence of blastocysts stained with anti-H3K27me3 (green) and anti-CDX2 (red) antibodies and DAPI (blue). The ratio of H3K27me3-positive-ICM blastocysts is shown in the right. Scale bar: 20 μm. Data are presented as mean values ± s.d. (**b**–**g**). Statistical significance was calculated with a two-tailed unpaired *t* test (**b**–**g**). n.s. not significant. Experiments were repeated three times independently; data of one representative experiment are shown (**h**). Source data are provided as a Source Data file.

ratio of GFP + /DsRed+ cells. The HEK293 cells were transfected with plasmids via the polyethyleneimine (PEI) transfection method.

To analyze the HR efficiency in mESCs, an I-SceI linearized HR (0.5 µg) cassette, together with a pCAG-DsRed vector (0.1 µg), was electroporated into $2 \times 10^5$ mESCs treated with DMSO or farrerol for the indicated times using the CG104 program on a Lonza 4D machine (Lonza, Cologne, Germany). Forty-eight hours post transfection, the cells were harvested for FACS analysis in a BD FACSVerse flow cytometer (BD Biosciences).

The HEK293 cell line was obtained from ATCC. The HCA2-hTERT cell line was obtained from the Gorbunova laboratory[66]. All cells included in the study were confirmed to be free of mycoplasma.

### Reagents and plasmids

Farrerol was purchased from Sigma (SML1389). RS-1 was purchased from ApexBio (C3357). Mirin was purchased from MCE (HY-19959). Primary antibodies used in this study were as follows: β-TUBULIN (Bioworld, Cat. # AP0064, WB: 1:2000). UCHL3 (Proteintech, Cat. # 12384-1-AP, WB: 1000), HA (Cell Signaling Technology, Cat. # 2367, WB: 1:1000), HA (Cell Signaling Technology, Cat. # C29F4, IF: 1:800), Flag (Abclonal, Cat. # AE005, WB: 1:5000), RAD51 (Abcam, Cat. # ab176458, WB: 1:10,000, IF: 1:1000 for cells, IF: 1:200 for embryos), GFP (Abclonal, Cat. # AE012, WB: 1:5000), γH2AX (Cell Signaling Technology, Cat. # 9718S, IF: 1:500), CDX2 (BioGenex, Cat. # MU392A-5UC, IF: 1:200), NANOG (Abcam, Cat. # ab80892, IF: 1:200), H3K27me3 (Diagenode, Cat. # C15410195, IF: 1:200), GATA6 (R&D Systems, Cat. # AF1700-SP, IF: 1:200), 5-Methylcytosine (5-mC) (Active Motif, Cat. # 61255, IF: 1:200). Molecular Probes DAPI (4′,6 Diamidino 2 Phenylindole) (Thermo Scientific, Cat. # D3571, IF: 1:1000). The following secondary antibodies were used: donkey anti-Rabbit 488 (Invitrogen, Cat. # A21206, IF: 2 µg/mL) and donkey anti-mouse 594 (Fisher Scientific, Cat. # A21203, IF: 4 µg/mL).

The sequences of siUCHL3 used in this study are as follows:
UCHL3-1: 5′-GGCACCAAGUAUAGAUGATT-3′;
UCHL3-2: 5′-UAGAAGUUUGCAAGAAGUUTT-3′;
UCHL3-3: 5′-GGCAGUUUGUUGAUGUGUATT-3′ (mouse);
UCHL3-4: 5′-GAGUUAAGAUUUAAUGCAATT-3′ (mouse);
UCHL3-5: 5′-GGACAAGAUGUGACAUCAUTT-3′ (mouse).
The sequences of sgUCHL3 used in this study are as follows:
UCHL3-1: 5′-ACCTTGAAAAAATTCCTGG-3′ (human);
UCHL3-2: 5′-GAAGTATTCAGAACAGAAG-3′ (human);
UCHL3-3: 5′-TCGATCATTTCCACGTTAAC-3′ (mouse);
UCHL3-4: 5′-GGTACCATGCTAAGAAGTTC-3′ (mouse);
UCHL3-5: 5′-CAGTAACTGTCTGACGGCAA-3′ (mouse);
UCHL3-6: 5′-GATTAGTCCAATCGTTCCAC-3′ (mouse).
The UCHL3-knockout cells were generated by the CRISPR–Cas9 system.

### Reverse transcription and quantitative-PCR (RT−qPCR)

Total RNA from 30 two-cell or morula embryos was purified using TRIzol reagent (Takara) and reverse-transcribed using 5×All-In-One RT Master Mix (ABM, Cat. # G492) according to the manufacturer's recommendations. RT−qPCR was performed using SYBR Premix Ex Taq II (Takara, Cat. # RR820B), and signals were detected with an ABI7500 Real-Time PCR System (Applied Bio Systems). The cDNA was diluted 1:10 in nuclease-free ddH2O and then used in RT−qPCR analysis. The H2A family member Z gene (*H2afz*) was used as an endogenous control. The primers used were as follows: *Uchl3*-F: TCCATGCCATTGCGAACAAC, *Uchl3*-R: CTGACCTTCATGTG CACTGGT; *H2afz*-F: TTGCAGCTTGCTATACGTGGAGATG, *H2afz*-R: TGTTGTCCTTTCTTCCCGATCAGC.

### Cellular thermal shift assay (CETSA)

The CETSA was performed as previously reported for DNA-PKcs[67]. Briefly, HEK293 cells were pretreated with DMSO or farrerol for 24 h.

Then, the cells were collected and heated at 42–60 °C for 3 min, followed by three freezing–thawing cycles using liquid nitrogen. Finally, the supernatants were collected for further Western blot analysis after centrifugation at 15,000×*g* for 15 min at 4 °C.

### Ub-AMC hydrolysis assay

The Ub-AMC (Ubiquitin C-terminal tagged 7-amido-4-methylcoumarin) and AMC used for hydrolysis assays was purchased respectively from Boston Biochem (Bio-Techne, R&D system, Cat. # U-550) and Aladdin (Cat. # A109916). The reaction mixture contained 50 mM Tris-HCl (pH 7.5), 1 mM EDTA, 0.1% BSA, 5 mM ATP/MgCl2 (freshly prepared), 1 mM DTT (freshly prepared), 0.05% Tween-20, and 5 nM UCHL3 with 0.1 µM farrerol. UCHL3 and farrerol mixture was diluted in reaction buffer prior to the addition of the substrate with final concentrations typically in the range of 25–600 nM. Ub-AMC hydrolysis was measured at Ex345/Em445 on SpectraMax iD3 apparatus. The data were analyzed with GraphPad Prism 9.0.

### Comet assay

Cells were seeded and maintained for 48 h and then collected and resuspended in PBS at a concentration of $3 \times 10^5$ cells per milliliter. To collect the embryos, farrerol- or DMSO-treated SCNT and IVF two-cell embryos were digested with pronase E (Sigma) and then incubated in $Ca^{2+}$-free CZB for 3–5 min. The cells were scattered into single cells, washed twice with 0.5% BSA-PBS and transferred to lysis buffer. The alkaline comet assay was performed according to the manufacturer's instructions (Trevigen, Cat. # 4250-050-K). DNA damage was measured in terms of tail moments using software (casplab_1.2.3b2).

### Immunofluorescence assay

Cells were seeded on coverslips in 12-well plates, treated with 2 Gy or 4 Gy X-ray and harvested at the indicated times. The cells were washed with PBS and fixed with 4% paraformaldehyde for 20 min at room temperature. To collect the embryos, farrerol or DMSO-treated SCNT and IVF two-cell embryos or blastocysts were directly transferred to 4% paraformaldehyde for 20 min. Then, the cells were washed twice with TBST and permeabilized with 0.2% Triton X-100 in TBST for 10 min at room temperature. The cells were washed three times with TBST and blocked with 2% BSA for 1 h at room temperature. The cells were incubated with primary and secondary antibodies that were diluted in PBS with 2% BSA overnight (primary antibody) at 4 °C and for 1 h (secondary antibody) at room temperature. For the staining of RAD51, cytoskeleton (CSK) buffer (10 mM PIPES, 300 mM Sucrose, 100 mM NaCl, 3 mM MgCl₂, 1 mM EGTA) was freshly prepared and gently added to each sample, and incubated for 10 min at 4 °C before fixing the cells or embryos.

As for the staining of 5-Methylcytosine (5-mC) modifications on DNA, it was necessary to transfer embryos to HCl at a concentration of 4 M after TBST washing at the end of punching and incubate at 37 °C for 30 min. The cells were then washed three times with Tris-HCl (pH 7.5) and three times with TBST before blocking and antibody incubation. After staining with DAPI, the coverslips were mounted, and images were acquired with Leica TCS SP8 confocal laser microscopy system (Leica Microsystems, Buffalo Grove, IL).

### Coimmunoprecipitation

Cells transfected with the indicated plasmids were collected and lysed in RIPA buffer (50 mM Tris-HCl (pH 7.4), 1 mM EDTA, 150 mM NaCl, 1% Triton X-100, 1% sodium deoxycholate, 0.1% SDS) with protein inhibitor on ice for 30 min prior to sonication on ice at 10% duty for 2 min, and the cell lysate was then centrifuged at 18,000×*g* for 10 min at 4 °C. The supernatant was collected for preclearing with Protein A/G-agarose beads and IgG for 2 h at 4 °C and was then centrifuged at 100×*g* for 3 min at 4 °C. The supernatant was collected, and then antibodies were added. After being rotated for 12 h at 4 °C, Protein A/G-agarose beads

were added to the lysate and incubated with rotation at 4 °C for 2 h. Then, the cells were washed five times with RIPA buffer, 2× sample buffer was added, and the samples were boiled for 10 min prior to western blot analysis.

### In vivo denatured deubiquitination assay and in vitro deubiquitination assay

The specific protocol for the in vivo and in vitro deubiquitination assays was performed as previously reported[28]. For the in vivo deubiquitination assay, control HEK293 cells and UCHL3 knockdown cells were treated with different concentrations of farrerol for 24 h and then collected, lysed, and centrifuged. The cell extracts were used to perform deubiquitination assays and immunoprecipitation experiments. For the in vitro deubiquitination assay, 10 μg HA-Ub and 5 μg RAD51-GFP plasmids were cotransfected into HEK293 cells via the poly-ethylenimine (PEI) transfection method. At 48 h post transfection, the cells were harvested, and ubiquitinated proteins were purified from the cell extracts with a GFP trap (Chromotek, Cat. # gta-20) under denaturing conditions. The Ub-RAD51 and UCHL3-GFP proteins were purified according to a standard protocol. Ubiquitinated proteins were incubated with recombinant UCHL3-GFP in deubiquitination buffer (50 mM Tris-HCl at pH 8.0, 50 mM NaCl, 1 mM EDTA, 10 mM DTT, 5% glycerol) for 4 h at 30 °C.

The Ub-CHOP2 assay was purchased from LifeSensors (PR1101). The assay was performed according to the manufacturer's instructions that were provided with the kit.

### Mice

Specific pathogen-free (SPF) grade mice, including BDF1, PWK, DBA2, and C57BL/6n mice, were housed in the animal facility at Tongji University, Shanghai, China. Mice were housed in individually ventilated cages with an environment with temperature ranging from 23 to 27 °C, humidity of 30–45%, and a light cycle with 12 h of light and 12 h of darkness. All mice were provided with free access to food and water. All experiments were performed in accordance with the University of Health Guide for the Care and Use of Laboratory Animals and were approved by the Biological Research Ethics Committee of Tongji University (TJAB04021104).

### Embryo collection and in vitro fertilization

The embryo collection and in vitro fertilization procedures were performed as previously described[68]. MII oocytes were obtained from BDF1 (C57BL/6n×DBA2) mice, which were superovulated by an injection of six IU pregnant mare serum gonadotropin (PMSG), followed by an injection of 6 IU of human chorionic gonadotropin (hCG) (San-Sheng Pharmaceutical) 48 h later. For in vitro fertilization, sperm were collected from the cauda epididymis of adult PWK/PhJ male mice (for RNA sequencing samples) or BDF1 male mice (for statistical development samples) and were incubated in G-IVF PLUS medium for 30 min at 37 °C under 5% CO2 to induce sperm swim-up. The zona pellucida of each MII oocyte was punched to create a small hole using a Piezo-drill micromanipulator. Oocytes were then placed in the prepared sperm suspension in G-IVF PLUS medium (Vitrolife). At 4 h post fertilization (hpf), these embryos were washed and cultured in G-1 PLUS medium. The fertilized oocytes that had developed to the two-cell stage by the next day were isolated for further culture and collection.

### SCNT embryo and small molecule treatment

BDF1 (C57BL/6n×DBA2) mice were used as oocyte donors, and cumulus cells from BDF1 or BPF1 (C57BL/6n×PWK/PhJ) female mice were used as nuclear donors. SCNT was performed as previously described[17]. In brief, MII oocytes were collected from superovulated enucleated oocytes at 14 h after hCG injection in a Chatot–Ziomek–Bavister (CZB) medium containing 5 mg/mL cyto-chalasin B (CB) by a Piezo-driven pipette (PrimeT 130 ech) with an

Olympus inverted microscope. For nuclear transfer, cumulus cells were injected into enucleated oocytes. The reconstructed oocytes were then incubated in CZB medium for 1 h before activation. Activation was performed by incubation in $Ca^{2+}$-free CZB medium containing 1 mM $SrCl_2$ and 5 mg/mL CB for 6 h, and then the oocytes were thoroughly washed and cultured in G-1 PLUS medium (Vitrolife) at 37 °C under 5% $CO_2$.

For small molecule treatment, the reconstructed oocytes were cultured in G-1-PLUS that contained molecules for 16 h post activation (hpa) or for the whole development process as indicated. The small molecules used include farrerol, RS-1 and mirin, and the concentrations used are described in Supplementary Table 1. DMSO (1000×) was used as the control treatment.

### In vitro synthesis and microinjection of mRNA

Mouse ESCs' cDNA was used as a template to amplify full-length mouse *h3f3a* and *h3f3b* products for in vitro transcription. T7-driven promoter and HA tag were added into the fragment by using the primer sequences as follows: *h3f3a*-T7-F: TAATACGACTCACTATAGGGatggcccgaaccaagcaga, *h3f3a*-HA-R: ttaAGCGTAATCTGGTACGTCG-TATGGGTAagcacgttctccgcgtatg, *h3f3b*-T7-F: TAATACGACTCACTA-TAGGGatggcccgaaccaagcaga, *h3f3b*-HA-R: ttaAGCGTAATCTGGTACG TCGTATGGGTAagctctctcccccccgtat.

Mouse *h3f3a* and *h3f3b* mRNA were synthesized with the mMES-SAGE mMACHINE T7 ULTRA Transcription Kit according to the manufacturer's instructions (Invitrogen, Cat. # AM1345). RNA was purified by phenol–chloroform extraction and ethyl alcohol precipitation. The integrity of manufactured mRNA was confirmed by electrophoresis. The precipitated RNA was dissolved in $H_2O$ at several concentration and stored at 80 °C until use. For mRNA injection, the enucleated oocytes or MII embryos were injected with ~10 pL of H3.3-HA mRNA (100 ng/μL) into the cytoplasm using a Piezo-driven micromanipulator.

### Depletion of *Uchl3* in SCNT embryos and in vitro fertilized embryos

siRNAs and sgRNAs against *Uchl3* were diluted in nuclease-free water. Cas9 mRNA and sgRNAs were produced as previously reported[36]. For the depletion of *Uchl3*, enucleated oocytes or MII embryos were injected with a mixture of 10 pL of siRNAs (20 μM), sgRNAs (50 ng/μL) and Cas9 mRNA (100 ng/μL), which targets *Uchl3*, by using a Piezo-driven micromanipulator. For nuclear transfer, after injection and incubating for 30 min in CZB medium, the nuclei of donor cells were injected into these pretreated enucleated oocytes. Reconstructed oocytes were then cultured in CZB medium for 1 h and further activated by 6 h of incubation in 1 mM $SrCl_2$ in $Ca^{2+}$-free CZB and 5 mg/mL cytochalasin B. Then, the SCNT embryos or zygotes were thoroughly washed and cultured in G-1 PLUS medium with or without farrerol treatment at 37 °C under 5% $CO_2$ to record further development or for embryo transfer.

### Embryo transfer and development rate record

DMSO-treated, farrerol-treated or *Uchl3*-depleted SCNT two-cell embryos were transferred into the oviducts of embryonic day 0.5 (E0.5) pseudopregnant female ICR mice. Cesarean section was performed at E19.5, and the surviving pups were fostered by lactating ICR females. The pups were photographed, weighed, and recorded to determine if they could respire autonomously. The placenta of the E19.5 pup was also weighed and recorded.

### Derivation and maintenance of mESCs

Wild-type mouse embryonic stem cells (mESCs) and SCNT-mESCs were generated as previously described[36]. IVF or SCNT E3.5 blastocysts were individually plated in each well of 96-well plates, which were coated with feeders (mitomycin C-treated MEFs) and further cultured

for the expansion of outgrowth. After 6–8 days, the cells were dissociated using 0.25% trypsin-EDTA (TE, 25200056, Thermo Fisher Scientific) and passaged into 48-well plates (p1, passage 1), followed by a second passage into 24-well plates (p2) and a third passage into 6-well plates (p3). The established mESC lines at p3 were genotyped to determine sex. The primers used were as follows: *sry*-F: CGTGGTGA-GAGGCACAAGTT, *sry*-R: AGGCAACTGCAGGCTGTAAA; *uty*-F: GAGTTCTTCTTGCGTTCACCATCTG, *uty*-R: CTATCTAATCCACAAAGCGCCTTCTTC.

All embryos and cells were cultured at 37 °C with 5% $CO_2$. mESCs were established and maintained on feeders in canonical serum-containing medium including knockout DMEM (Gibco, Cat. # 10829) with 15% fetal bovine serum (FBS, Gibco, Cat. # 16000-044), 1 mmol/L L-glutamine (Thermo Fisher Scientific, Cat. # 25030164), 100× nucleosides (Sigma-Aldrich, Cat. # M6250), 100× NEAA (Millipore, Cat. # TMS-001), 0.11 mmol/L 2-mercaptoethanol (Invitrogen, Cat. #21985023), $10^3$ U/mL LIF (Millipore, Cat. # ESG1107), 100× penicillin/streptomycin (Gibco, Cat. # 15140-122), 1 mM PD0325901 (Selleck, Cat. #S1036) and 3 mM CHIR99021 (Selleck, Cat. # S2924).

### Karyotyping analysis and detection of chromosome copy number variation

The karyotype analysis was described in a previous report[36]. Briefly, the mESCs were cultured in a corresponding medium with 0.25 µg/mL colcemid (Invitrogen, Thermo Fisher Scientific) for 2.5 h and dissociated and collected with 0.05% trypsin-EDTA. Then, we incubated the cells in a hypotonic solution containing 0.4% potassium chloride and 0.4% sodium citrate at 37 °C for 5 min. To analyze the embryonic chromosomes, morulae were transferred to CZB containing 0.25 µg/mL colcemid and incubated for 3 h and then transferred to a 1:1 mixture of 1% sodium citrate and 30% FBS hypotonic solution for 5–7 min at room temperature. After fixing, the cells were mounted on coverslips, left to dry, and then stained with Giemsa at 37 °C for 15 min. The chromosome number of the separated nucleus was counted under an inverted microscope (Leica). For each sample, at least 20 nuclei were analyzed.

For the analysis of chromosome copy number variation, the four-cell embryo was dispersed into a single blastomere after digesting the zona pellucida with 0.5% pronase E. Subsequently, one or two blastomeres were collected and placed in 5 µL preservation solution (Yikon Genomics, Cat. # XK-002), frozen at −80 °C, or immediately transported to Yikon Genomics Co., Ltd. on dry ice for detection of chromosome copy number variation.

### Total RNA library construction and sequencing

The zona pellucidae of the embryos were removed with 0.5% pronase E, and the embryos were then incubated in $Ca^{2+}$-free CZB medium for 10 min. Polar bodies were removed by gentle pipetting using a fire-polished glass needle. Morula-stage IVF embryos and SCNT embryos were collected at 67–68 h.p.f. and h.p.a., respectively.

Harvested blastomeres and extraembryonic ectoderm cells of E7.5 embryos were placed in 500 µL TRIzol reagent (Takara), and total RNA was isolated by chloroform extraction coupled with isopropanol precipitation. The RNA was then washed twice with 75% ethanol and eluted with nuclease-free water (Invitrogen). Illumina-compatible RNA-seq library preparation was generated by SMARTer Stranded Total RNA-Seq Kit v2 following the manufacturer's instructions (Takara, Cat. # 634412). Briefly, 10 ng of total RNA was converted to cDNA, and then adapters for Illumina sequencing (with specific barcodes) were added through PCR using only a limited number of cycles. After purifying the PCR products with depleted ribosomal cDNA, the cDNA fragments were further amplified with primers that were universal to all libraries. Sequencing was performed by Berry Genomics Co., Ltd. using the NovaSeq system developed by Illumina. Two or three biological replicates were analyzed under each treatment condition.

### RNA-seq analysis

Sequencing reads were first trimmed to remove adapters and then mapped to the annotated mouse reference genome (UCSC mm10, https://genome.ucsc.edu/) using HISAT2 (v2.2.1) and sorted by SAMtools (v1.12). The reads were counted by TEcount (v2.2.1). The counts were normalized using variance stabilizing transformations methods by DESeq2 (v1.24.0) and subsequently used for detecting differentially expressed genes (DEGs). The gene expression level was quantified as fragments per kilobase per million mapped reads (FPKM) using StringTie (v2.1.7) and the R package Ballgown (v2.16.0).

### Reporting summary

Further information on research design is available in the Nature Portfolio Reporting Summary linked to this article.

## Data availability

The RNA-seq data generated in this study have been deposited in the Genome Sequence Archive in National Genomics Data Center, China National Center for Bioinformation/Beijing Institute of Genomics, Chinese Academy of Sciences, under accession code CRA007477. The target gene list of CDX2 and NANOG stems from CHEA Transcription Factor Targets, which can be accessed through Harmonizome. The mass spectrometry proteomics data (including raw MS data) in this study have been deposited to the ProteomeXchange Consortium via the PRIDE partner repository with the dataset identifier PXD036308. The UCHL3–farrerol complex structure used herein is available in the PDB database under accession code 7YV4. Source data are provided with this paper.

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

## Acknowledgements

We thank the Tsinghua University Branch of China National Center for Protein Sciences (Beijing) and Tsinghua University Technology Center for Protein Research for the X-ray crystallography facility support. The work was supported by grants from the Chinese National Program on the Key Basic Research Project (Grant Nos. 2021YFA1102000 to Z.M., 2021YFA1100300 and 2020YFA0112500 to J.C.), the National Natural Science Foundation of China (Grant Nos. 82071565 to Z.M., 31721003 and 31820103009 to S.G., 81972457 and 32171288 to Y.J., 32200603 to W.Z., 81973500 to X.X., 32270856 and 32070857 to J.C. and 32100652 to G.Y.), the Key Project of the Science and Technology of Shanghai Municipality (Grant Nos. 19JC1415300 to J.C.), the Shanghai Sailing Program (Grant Nos. 22YF1434500 to W.Z.), the Open Project Program of State Key Laboratory of Natural Medicines (Grant Nos. SKLNMKF202105 to Z.M.), the Shanghai "Super Postdoctoral" incentive program (Grant Nos. 2022572 to M.W.) and the China Postdoctoral Science Foundation (Grant Nos. 2022M722421 to M.W., 2021M692438 and 2022T150482 to G.Y.).

## Author contributions

Y.J., Sha.G., X.X., and Z.M. designed and conceived the study. J.C. designed the SCNT-related experiments. Wein.Z., M.W., Z.S., and Q.F. performed most of the experiments. Q.F. did most of the computational data analysis with the help of Shu.G. and G.Y. Weit.Z. and H.T. contributed by performing the protein crystallography and data analysis. X.S. and J.Y. carried out coimmunoprecipitation and Western blot assays. Z.S., Ha.W. contributed by performing the mutant constructions and protein purifications. M.W., J.C., Q.Z., X.K., and Ho.W. performed the SCNT assay and data analysis. Wein.Z., M.W., Z.S., Q.F., J.C., Sha.G., and Z.M. wrote the manuscript. J.C., Z.M., and Sha.G. helped with figures, data analysis, and experimental design. All authors discussed and interpreted the data and approved the manuscript.

## Competing interests

The authors declare no competing interests.
