## [Peer Review File · Nature Communications]

Farrerol directly activates the deubiquitinase UCHL3 to promote DNA repair and reprogramming when mediated by somatic cell nuclear transferREVIEWER COMMENTS

Reviewer #1 (Remarks to the Author):

In this study Zhang and colleagues showed that deubiquitinase UCHL3's specific role in farrerol-mediated homologous recombination (HR) is essential for DNA repair and reprogramming. To demonstrate this point, the authors used LiP-SMap and BIAcore assay to found out specific protein deubiquitinase UCHL3, which deubiquitinate and activate RAD51 to regulate HR under farrerol condition. Also identified two amino acid residues, K187 and R215, which are critical to direct binding of farrerol to UCHL3. The authors found farrerol treatment can increase somatic cell nuclear transfer (SCNT) efficiency by enhancing HR repair. The authors provide a very extensive set of data based on different assays and disciplines. It is an impressive study that will inspire new research in all of these fields. Overall, the reviewer believes it will be suitable for Nat Comms upon meaningful revision.

There are several questions or concerns for the manuscript.

1). Figure 3d-e. The DNA damage level (number of γ H2AX foci) is needed to show. It is possible that the recruitment time is affected by the dose of ionizing irradiation (IR). Recommend more than a single dose to determine if this holds true at more than a single level of DNA damage.

2). DNA damage after SCNT should be measured in γ H2AX foci not only at the blastocyst stage, but primarily at the cleavage stage. And intensity is not a good measure, which may be a reflection of cell cycle stage. A consistent time point relative to the cell cycle should be evaluated. Blastocyst is too far downstream of the reprogramming. Fig. S3C, intensity is again not a good measure for Rad51, it should be the number of foci.

3) In figure 4, needs to show IVF control groups with and without treatment essentially throughout the Figure, all panels. Some of the info is split into supplemental Figures, such as in Fig. S7, but this is difficult for the reader to compare and evaluate.

4) the evaluation of karyotypes through spreads is not a strong assay and vulnerable to errors, losses. IN addition to sequencing based methods, an alternative is to evaluate chromosome segregation at anaphase or frequency micronucleation. At least one complementary assay should be added given that weakness.

5). UCHL3 also has function in single-strands DNA break repair. Stimulation of HR is not the sole interpretation and should be discussed appropriately at the minimum. If you can perform Two-tailed-Comet assay to distinguish single-stranded DNA and double-stranded DNA damage instead of alkaline comet assay, would be more better to support the main UCHL3 function in the paper. Alternative is a detailed quantification of γ H2Ax foci at a specific time point.

Minor comments

Figure 2d and f. please add ratio of UCHL3/ACTIN in graphs.

Figure 2 misspells temperature

The sentence "The in vitro biochemical assay revealed that farrerol at a concentration of 0.1 μ M significantly increased the deubiquitinase activity of UCHL3" does not match Fig. 3h, are you mean fig. 3j?

The sentence "Transient treatment with mirin, an inhibitor of MRE11 that is essential for the initiation of HR repair, severely impaired chromosomal integrity, presenting only 20% of euploid karyotypes",

you did not mention which figure, and please add "(fig. 4e)".

The sentence "In addition, more ICM numbers and recovered TE/ICM ratios were observed in the farrerol-treated SCNT blastocysts, which indicated that the first lineage differentiation potential in the SCNT embryos was superior after the addition of farrerol addition". Please remove last "addition".

References need to be accurately cited for what they show. Line 117. Mizutani shows that cloned embryos show chromosome segregation errors and that this is an obstacle to development. They provide no mechanism. Replication dependent DNA damage is shown in Reference 2, but not in Ref 20.

Also in line 264, references are lumped together that show very different things. Not clarified what ref 47 contributes there.

The introduction has a lot of associative text, like the lifespan of whales. This is not supporting the strength of the paper and instead weakens it.

Mirin treatment. The fact that inhibition impairs development is not in question. Mirin inhibition dramatically affects DNA repair during normal DNA replication. The question is whether it affects nuclear transfer embryos MORE than IVF embryos. Treatment of IVF embryos as a control is not shown.

Fig. S6C claims significant differences of placental weights between SCNT conditions, but it is not convincing.

Reviewer #2 (Remarks to the Author):

In this manuscript, Zhang et al. describes the discovery of UCHL3 as the target of a natural medicinal compound, farrerol, which has been previously shown to enhance HR repair in CRISPR experiments. Farrerol appears to bind to UCHL3 with high affinity and stimulate its catalytic activity. Importantly, the stimulatory effect of farrerol on HR is abrogated when UCHL3 is knocked-down or knocked-out. Interestingly, farrerol significantly enhances the efficiency of SCNT, which has low success rate in part due to genomic instability. The authors argue that farrerol acts through UCHL3 in this process to overcome reprogramming barriers. These discoveries provide significant mechanistic insights into the medicinal effects of a widely used natural compound.

The authors presented compelling evidence that UCHL3 is a target of farrerol, however, how UCHL3 activity is affected by farrerol is less clear. In addition, although the evidence that farrerol enhances SCNT success rate is comprehensive, UCHL3 being the main target in this process is not well supported. Below are the major points that need to be addressed:

1. The co-crystal structure of farrerol and UCHL3 is very compelling. However, the authors did not present an overlay of their structure with the structure of UCHL3 in complex with ubiquitin (1XD3). Structurally it remains unclear whether farrerol affects ubiquitin binding by UCHL3.
2. The effects of R215A and K187A mutations on farrerol binding should be quantified by SPR.
3. Mechanistically how farrerol enhances UCHL3 activity is not addressed. Is the effect on K_m or K_{cat} of the enzyme? Because the UCH-family of DUBs are regulated by the active-site crossover loop, one should compare a minimal substrate such as Ub-AMC with a bulky substrate used in Ub-CHOP reporter (Fig. 3h).
4. As presented, the assays in Fig. 3i and 3j are not convincing. The Ub signals that appear in Rad51 immunoprecipitants may not be attached to Rad51, rather they could be attached to other Rad51-associated proteins. In Fig. 3j, while the HA-Ub signal decreases upon incubation with UCHL3, there is no accompanying increased signal of Rad51-GFP, further arguing that those Ub are not attached to

Rad51.

5. Fig. 2d-f shows beautifully that in the absence of UCHL3, farrerol has no effect on Rad51 foci. However, these experiments cannot exclude that farrerol acts on a factor that's downstream of UCHL3 in the HR pathway. To address this issue, the authors should rescue the UCHL3 knockout with either wildtype or non-binding mutant of UCHL3, such as R215A or K187A.

6. Given that farrerol enhances UCHL3 activity about two-fold (Fig. 3h), one would expect that overexpression of UCHL3 is sufficient to enhance HR. Has this been tested?

7. The interpretation of Fig. 4i is problematic. Since UCHL3 knockout did not provide any viable E19.5 pups by SCNT, one cannot conclude that farrerol acts through UCHL3. There are many genes whose knockout could lead to the lethality phenotype and they are not farrerol targets.

8. The effects of farrerol on H3K27Me3 and transcription programs argue against the author's conclusion that UCHL3 is the main target of farrerol in SCNT. Given that farrerol has other well-documented activities outside HR, it seems more likely that farrerol is acting through multiple mechanisms in enhancing SCNT efficiency.

9. Line 426: "compared with the rough regulation pattern of RAD51", please clarify what the authors mean by "rough regulation pattern".

Reviewer #3 (Remarks to the Author):

The paper by W. Zhang et al. reports that farrerol, a natural flavanone, directly targets a deubiquitinase UCHL3, which activates a recombinase RAD51, thus promoting homologous recombination (HR) DNA repair. The authors' precise LiP-SMap combined with a BIAcore assay as well as cocrystallization experiments identified direct binding of farrerol to UCHL3 through specific amino acid residues. Finally, the authors demonstrated that the birth rate of SCNT was significantly improved from < 1% to about 4% by treatment with farrerol. RNA-seq analysis revealed that this improvement could be attributable to corrected expression profiles of developmentally important genes and repression of somatic cell-related genes. The experiments were well-designed, the results were clearly presented, and final conclusions were logically drawn from the experimental results obtained. Overall, the paper is of a highly scientific quality, and will attract broad interests of researchers in the cell biology as well as developmental /reproductive biology. However, there are some points to be improved so that the paper can be a more convincing paper in the related fields.

Major points

1. The most striking finding in this study is the advantageous changes in the transcriptomic feature of SCNT embryos following farrerol treatment. As far as its chemical and biochemical backgrounds are concerned, farrerol is involved in DNA repair at the time of genomic reprogramming, not in genomic (epigenetic) reprogramming per se. Although the authors mentioned that it is an interesting topic for future research, I strongly recommend the authors to perform additional experiments that may help the authors propose some possible mechanism, because this could be the central notion of this study. For example, if the farrerol treatment induces open chromatin in reconstructed oocytes, this might offer a reasonable scenario for efficient SCNT (like TSA treatment). ATAC-seq or similar analysis may work well. Alternative ways may include observation of the level of maternal H3.3 incorporation into the donor nucleus in the reconstructed oocytes, if the authors easily prepare H3.3 mRNA for injection.

2. Another intriguing finding is the lack of the positive effect of RS-1 (RAD51 activator) treatment on SCNT, although farrerol promotes RAD51-mediated DNA repair as well. Lee et al. (ref. 22) reported that the mouse cloning efficiency (birth rate of cloned mice) increased from 0.4% to 5.2% by RS-1 treatment. In vitro development of SCNT embryos was also significantly improved; from 35% to 66% blastocyst rate. In this study, the blastocyst rate was about 40% following RS-1 treatment, showing no improvement from the control SCNT level. By what reason do the authors assume these two different outcomes occurred? Why RS-1 did not have any positive effect on SCNT in this study? In the discussion, the authors mentioned that "farrerol functions in a more refined way", but this is not a

scientific explanation.

3. The authors argued that placental hyperplasia associated with mouse SCNT was alleviated by farrerol treatment. At least three SCNT papers so far have reported that loss of maternal noncanonical imprinting was responsible for this placental hyperplasia. As the authors produced cloned mice with PWKxB6 donors, the allelic data on the noncanonical imprinting status of the placentas of farrerol-treated SCNT should be presented.

4. As shown in Fig. 4d and Extended Data Fig. 3b, farrerol exerted its effect on the 2-cell to 4-cell transition, indicating that farrerol improved zygotic genome activation. However, the authors did not present any transcriptome data that support this. It is important to see if the farrerol treatment can activate the reprogramming resistant regions (RRRs) repressed by H3K9me3 in SCNT embryos (Matoba et al. Cell 2014).

Minor points

1. Abstract: In the last sentence, there is: "provided a more feasible method to promote SCNT efficiency when compared to other methods". I wonder what data in this study guarantee "more feasible than other methods". The authors would better explain this.

2. Line 104: ref. 18 is not a SCNT paper.

3. Lines 111-113: "As a comparison....". Appropriate reference(s) should be cited here.

4. Line 275: Fig. 4d -> Fig. 4d-e

5. Line 275: Extended Data Fig. 3a shows IVF embryo development, not SCNT.

6. Line 363: The experimental conditions of Uchl3-sg/siRNA were different between SCNT and IVF; oocytes were used for the former, but zygotes were used for the latter. It is essential to inject sg/siRNA into oocytes before IVF to see the effect of sg/siRNA on normal embryonic development more correctly.

7. Line 367: Fig. 4j -> Fig. 4i-j

8. Discussion: In the Discussion section, it may be more natural to discuss first about UCHL3 as a target of farrerol, in line with the order of the Results section. For example, the paragraph "Previous studies have revealed that farrerol exhibits..." (Line 465) can be the first.

Responses to reviewer's comment

Reviewer #1 (Remarks to the Author):

In this study Zhang and colleagues showed that deubiquitinase UCHL3's specific role in farrerol-mediated homologous recombination (HR) is essential for DNA repair and reprogramming. To demonstrate this point, the authors used LiP-SMap and BIAcore assay to found out specific protein deubiquitinase UCHL3, which deubiquitinate and activate RAD51 to regulate HR under farrerol condition. Also identified two amino acid residues, K187 and R215, which are critical to direct binding of farrerol to UCHL3. The authors found farrerol treatment can increase somatic cell nuclear transfer (SCNT) efficiency by enhancing HR repair. The authors provide a very extensive set of data based on different assays and disciplines. It is an impressive study that will inspire new research in all of these fields. Overall, the reviewer believes it will be suitable for Nat Comms upon meaningful revision.

We thank the reviewer for the positive comments.

There are several questions or concerns for the manuscript.

1). Figure 3d-e. The DNA damage level (number of γ H2AX foci) is needed to show. It is possible that the recruitment time is affected by the dose of ionizing irradiation (IR). Recommend more than a single dose to determine if this holds true at more than a single level of DNA damage.

We thank the reviewer for the valuable suggestion. As reviewer suggested, we performed immunofluorescence experiments to investigate the level of DNA damage upon ionizing irradiation at different doses including 2 Gy and 4 Gy. The results showed that farrerol promoted the recruitment of RAD51 at DNA damage sites (Fig. 3c, d) and accelerated γ H2AX clearance at 4 hours post IR at a dosage of 2 Gy (Fig. 3e, f). Similarly, upon IR at a dosage of 4 Gy, we found that farrerol treatment significantly promoted the RAD51 recruitment and accelerated γ H2AX clearance, and that depletion of UCHL3 abolished the farrerol-mediated effects (Supplementary Fig. 2e-h). Taken together, these data strongly indicate that farrerol promotes HR repair to stabilize genomes by facilitating RAD51 recruitment. We have included the data in the new Figure 3e-f and Supplementary Figure 2e-h.

Fig. 3e

Fig. 3f

Fig. S2e**Fig. S2f****Fig. S2g****Fig. S2h**
2). DNA damage after SCNT should be measured in γ H2AX foci not only at the blastocyst stage, but primarily at the cleavage stage. And intensity is not a good measure, which may be a reflection of cell cycle stage. A consistent time point relative to the cell cycle should be evaluated. Blastocyst is too far downstream of the reprogramming. Fig. S3C, intensity is again not a good measure for Rad51, it should be the number of foci.

We thank the reviewer for the valuable suggestion. We performed immunofluorescence assay to measure the number of γ H2AX foci in embryos at the cleavage stage and blastocyst stage. The results clearly showed that the number of γ H2AX foci in SCNT embryos was much higher than that in IVF embryos, and the farrerol treatment greatly decreased the number of γ H2AX foci in SCNT embryos compared to that in non-treated SCNT embryos. We have included these data in new Figure 4d and Supplementary Figure 4d.

Fig. 4d**Fig. S4d**
In order to solve the staining problem of RAD51 in embryos, five RAD51 antibodies were tested, including ab176458 and ab63801 from abcam, ET1705-96 from huabio, PC130 from EMD Millipore, 14961 from Proteintech. As shown in the immunofluorescence staining images below, none of the five antibodies could clearly show foci in the nucleus of IVF embryos. Meanwhile, the overall cytoplasm was deeply stained (Fig. R-1). We speculated that this might be due to the existence of some new types of embryonic RAD51 or the unique epigenetic state in embryos affects the formation of RAD51 foci. Therefore, based on the staining results of RAD51, we still used intensity of fluorescence to measure RAD51 in both IVF and SCNT embryos. Moreover, among these antibodies, antibody PC130 has a slightly better staining effect and is the antibody used in the revised manuscript.

Only shown in the point-by-point response letter

Fig. R-1

3) In figure 4, needs to show IVF control groups with and without treatment essentially throughout the Figure, all panels. Some of the info is split into supplemental Figures, such as in Fig. S7, but this is difficult for the reader to compare and evaluate.

Thank the reviewer for the valuable suggestion. As the reviewer suggested, we have conducted assay to show IVF control groups with and without farrerol treatment throughout the Figure, all panels. In addition, we also adjusted the layout of the Figures. We have included relative data in new Figure 4c-f, 5e-i; Supplementary Figure 4a-f, 5d, e, 6a-e, 7a, c, d.

4) The evaluation of karyotypes through spreads is not a strong assay and vulnerable to errors, losses. In addition to sequencing based methods, an alternative is to evaluate chromosome segregation at anaphase or frequency micronucleation. At least one complementary assay should be added given that weakness.

We thank the reviewer for the valuable suggestion. We agree with the reviewer on the raised point. As the reviewer suggested, we performed chromosome copy number variation (CNV) analysis. 1-2 blastomeres in 4-cell embryos from IVF or SCNT group with and without farrerol treatment were collected for the analysis. Consistent with our karyotype analysis (Fig.4e), the data revealed that the farrerol treated SCNT embryos exhibited reduced CNV abnormality, and avoided the loss of certain large genome fragments in comparison to control SCNT embryos (Fig.4b and Supplementary Fig. 4a), indicating that farrerol could maintain the genome integrity during SCNT process. We have included this data in new Figure 4b and Supplementary Figure 4a.

Fig. 4b**Fig. S4a**
5). UCHL3 also has function in single-strands DNA break repair. Stimulation of HR is not the sole interpretation and should be discussed appropriately at the minimum. If you can perform Two-tailed-Comet assay to distinguish single-stranded DNA and double-stranded DNA damage instead of alkaline comet assay, would be more better to support the main UCHL3 function in the paper. Alternative is a detailed quantification of γ H2Ax foci at a specific time point.

We thank the reviewer for the valuable suggestion. As the reviewer suggested, we performed immunofluorescence experiments with an anti- γ H2AX antibody for measuring DNA DSBs in 2-cell stage SCNT embryos after *Uchl3* depletion with or without farrerol treatment. The result clearly showed that farrerol could reduce the number of DNA DSBs, whereas *Uchl3* depletion abrogated its effect (Fig. 6d). We have included this data in new Figure 6d.

Fig. 6d
Minor comments

1. Figure 2d and f. please add ratio of UCHL3/ACTIN in graphs.

We thank the reviewer for the suggestion. We have added the ratio of UCHL3/ACTIN in the graphs.

2. Figure 2 misspells temperature

We are sorry for our carelessness. We have corrected this error in the revised version.

3. The sentence ‘‘The *in vitro* biochemical assay revealed that farrerol at a concentration of 0.1 μ M significantly increased the deubiquitinase activity of UCHL3’’ does not match Fig. 3h, are you mean fig. 3j?

We are sorry for the confusion. The *in vitro* biochemical assay here refers to the Ub-CHOP2 assay, and the concentration of farrerol used in this experiment was 0.1 μ M.

4. The sentence ‘‘Transient treatment with mirin, an inhibitor of MRE11 that is essential for the initiation of HR repair, severely impaired chromosomal integrity, presenting only 20% of euploid karyotypes’’, you did not mention which figure, and please add ‘‘(fig. 4e)’’.

We are sorry for our carelessness. The mirin group in Fig. 4e of the original manuscript should have been IVF, not SCNT, which we corrected in the revised version. The description of this part has been adjusted, and the modified content is as follows: The transient treatment with mirin (~16 h), an inhibitor of MRE11 that is essential for the initiation of HR repair, significantly blocked SCNT mediated reprogramming and retarded early embryonic development (Fig. 4g, h and Supplementary Table 2).

5. The sentence ‘‘In addition, more ICM numbers and recovered TE/ICM ratios were observed in the farrerol-treated SCNT blastocysts, which indicated that the first lineage differentiation potential in the SCNT embryos was superior after the addition of farrerol addition’’. Please remove last ‘‘addition’’.

We have removed last ‘‘addition’’ in the revised version.

6. References need to be accurately cited for what they show. Line 117. Mizutani shows that cloned embryos show chromosome segregation errors and that this is an obstacle to development. They provide no mechanism. Replication dependent DNA damage is shown in Reference 2, but not in Ref 20.

Also in line 264, references are lumped together that show very different things. Not clarified what ref 47 contributes there.

We thank the reviewer for pointing out the issues. We have made corrections accordingly.

7. The introduction has a lot of associative text, like the lifespan of whales. This is not supporting the strength of the paper and instead weakens it.

We thank the reviewer for the suggestion. We removed this sentence in the revised version. We also modified the introduction part to make it more concise and clearer.

8. Mirin treatment. The fact that inhibition impairs development is not in question. Mirin inhibition dramatically affects DNA repair during normal DNA replication. The question is whether it affects nuclear transfer embryos MORE than IVF embryos. Treatment of IVF embryos as a control is not shown.

We thank the reviewer for the valuable suggestion. The treatment of Mirin in IVF embryos was performed and the data was shown in Supplementary Table 2 and Fig. R-2. We did find that Mirin affected SCNT embryos more severely than IVF embryos.

In the condition of long-term treatment (1-cell to blastocyst, indicated as Mirin-maintain), only 63.05±6.23% IVF embryos could normally develop to 2-cell stage. But eventually, all these 2-cell embryos were arrested at 2-4 cell stage and no blastocyst could be obtained. As a comparison, all the SCNT embryos were blocked in 1-cell stage under such a long-term Mirin treatment condition.

In a 16 h treatment condition (indicated as Mirin-16 h), we found that the retarded IVF embryos could resume development when Mirin was removed. However, the blastocyst rate (81.78±2.83%) was lower than that in the control group (93.44±1.23%). As a comparison, although the SCNT embryos could resume to develop beyond the 2-cell stage, most of these embryos were blocked at the 4-cell stage.

These data indicated that the recovery ability or stress resistance of IVF embryos is significantly stronger than that of SCNT embryos during Mirin treatment. Mirin has a greater effect on the development of SCNT embryos than IVF embryos. We have included related data in Supplementary Table 2.

Table. S2

Preimplantation Development of IVF and SCNT embryos							
Embryo type	mRNA injection	Treatment	No. of replicates	No. of reconstructed 2-cell embryos	%4-cell per 2-cell ±SD	%morula per 2-cell ±SD	%blastocyst per 2-cell ±SD
IVF	/	Control-DMSO (1000×, maintain)	3	60	100±0.00	96.11±2.83	93.44±1.23
	/	Mirin (50 µM, maintain)	4	58	0	0	0
	/	Mirin (50 µM, 16 h)	2	45	96.15±3.85	92.31±7.69	81.78±2.83*
	/	Farrerol (0.05 µM, maintain)	3	65	100±0.00	98.25±2.48	92.45±1.62
	/	Farrerol (0.05 µM, 16 h)	2	40	100±0.00	100±0.00	100±0.00**
SCNT	/	Control-DMSO (1000×, maintain or 16 h)	16	346	70.16±7.28	50.80±9.19	37.69±7.62
	/	Mirin (50 µM, maintain)	4	0	0	0	0
	/	Mirin (50 µM, 16 h)	3	53	23.21±11.04***	13.84±9.00***	7.14±6.73***
	/	Farrerol (0.05 µM, maintain)	2	38	63.06±1.94	0	0
	/	Farrerol (0.05 µM, 16 h)	10	217	89.95±5.20***	81.09±10.80***	70.02±10.00***
	/	Farrerol (0.1 µM, 16 h)	6	148	85.83±6.29***	74.50±7.57***	53.86±7.78***
	/	Farrerol (0.2 µM, 16 h)	3	58	41.55±2.62***	39.79±3.33	32.92±3.39
	/	RS-1 (5 µM, 16 h)	2	57	72.95±5.31	52.56±0.38	33.63±7.54
	/	RS-1 (10 µM, 16 h)	5	117	80.53±10.37*	67.61±16.87*	46.67±12.78
	/	RS-1 (10 µM, 22 h)	2	44	91.37±3.37**	77.47±1.47**	63.58±0.42***
IVF	/	RS-1 (15 µM, 16 h)	2	53	63.65±5.32	41.88±3.95	29.38±8.55
	si/gControl	Control-DMSO (1000×, 16 h)	2	60	100±0.00	100±0.00	98.28±1.72
SCNT	siUch3 (20 µM)+sgUch3 (50 ng/µL)	DMSO (1000×, 16 h)	2	56	98.21±1.79	96.43±0.00	91.07±1.79
	si/gControl	Control-DMSO (1000×, 16 h)	3	59	69.59±2.98	50.79±6.54	42.36±0.35
SCNT	siUch3 (20 µM)+sgUch3 (50 ng/µL)	DMSO (1000×, 16 h)	3	58	65.33±6.23	41.92±8.43	40.69±7.67
	si/gControl	Farrerol (0.05 µM, 16 h)	2	39	90.08±4.37*	80.16±8.73*	69.44±2.78***
	siUch3 (20 µM)+sgUch3 (50 ng/µL)	Farrerol (0.05 µM, 16 h)	4	89	70.00±5.11	63.34±5.37	52.78±2.78**

IVF, *in vitro* fertilization; SCNT, somatic cell nuclear transfer

*P<0.05; **P<0.01; ***P<0.001 as compared with the corresponding control group.

Only shown in the point-by-point response letter

Fig. R-2

9. Fig. S6C claims significant differences of placental weights between SCNT conditions, but it is not convincing.

We thank the reviewer for the question. We rephrased the text as follows: More strikingly, farrerol treatment effectively recovered the persistent hyperplasia of SCNT placenta (Matoba, et al. 2018; Yang, et al. 2021). In particular, the weight of placentae produced by SCNT embryos was partially reduced by farrerol treatment (Supplementary Fig. 7c). Moreover, the structure of these placentae from farrerol-treated SCNT embryos was more organized, demonstrating three-layered smooth edges, which provided structural integrity (Supplementary Fig. 7d).

Fig. S7c

Fig. S7d

Reviewer #2 (Remarks to the Author):

In this manuscript, Zhang et al. describes the discovery of UCHL3 as the target of a natural medicinal compound, farrerol, which has been previously shown to enhance HR repair in CRISPR experiments. Farrerol appears to bind to UCHL3 with high affinity and stimulate its catalytic activity. Importantly, the stimulatory effect of farrerol on HR is abrogated when UCHL3 is knocked-down or knocked-out. Interestingly, farrerol significantly enhances the efficiency of SCNT, which has low success rate in part due to genomic instability. The authors argue that farrerol acts through UCHL3 in this process to overcome reprogramming barriers. These discoveries provide significant mechanistic insights into the medicinal effects of a widely used natural compound.

We thank the reviewer for the positive comments.

The authors presented compelling evidence that UCHL3 is a target of farrerol, however, how UCHL3 activity is affected by farrerol is less clear. In addition, although the evidence that farrerol enhances SCNT success rate is comprehensive, UCHL3 being the main target in this process is not well supported. Below are the major points that need to be addressed:

1. The co-crystal structure of farrerol and UCHL3 is very compelling. However, the authors did not present an overlay of their structure with the structure of UCHL3 in complex with ubiquitin (1XD3). Structurally it remains unclear whether farrerol affects ubiquitin binding by UCHL3.

We thank the reviewer for the valuable suggestion. we aligned the pre-deposited structure of UCHL3 in complex with ubiquitin (1XD3) with our cocrystal structure using US-align (Zhang, et al. 2022). The normalized TM-score of these two structures is 0.97 and structural superimposition was shown in Supplementary Fig. 1e. The alignment indicated that farrerol did not affect ubiquitin binding by UCHL3. We have included the data in new Supplementary Figure 1e.

Fig. S1e

2. The effects of R215A and K187A mutations on farrerol binding should be quantified by SPR.

We thank the reviewer for the insightful suggestion. As the reviewer suggested, we performed SPR assay with the purified UCHL3-2A mutant, which contains both K187A and R215A mutations. We found that mutating both R215 and K187 into alanine on UCHL3 abrogated its binding with farrerol, indicating that farrerol binds to UCHL3 through two amino acid residues. We have included the data in new Supplementary Figure 1f.

Fig. S1f

3. Mechanistically how farrerol enhances UCHL3 activity is not addressed. Is the effect on K_m or K_{cat} of the enzyme? Because the UCH-family of DUBs are regulated by the active-site crossover loop, one should compare a minimal substrate such as Ub-AMC with a bulky substrate used in Ub-CHOP reporter (Fig. 3h).

We thank the reviewer for the suggestion. We agree with the reviewer on the issue. As the reviewer suggested, we have performed Ub-AMC assay for examining the DUB activity of UCHL3 in the presence or absence of farrerol. Michaelis-Menten plot for analysis confirmed that farrerol affects both K_m and K_{cat} , with a slight increase in K_m from 47.72 to 56.27 nM and a drastic increase in K_{cat} from 11.99 to 19.64 s⁻¹ (Fig. 3i). This result indicates that farrerol preferentially activates UCHL3 enzymatic activity. We have included the data in new Figure 3i.

Fig. 3i

4. As presented, the assays in Fig. 3i and 3j are not convincing. The Ub signals that

appear in Rad51 immunoprecipitants may not be attached to Rad51, rather they could be attached to other Rad51-associated proteins. In Fig. 3j, while the HA-Ub signal decreases upon incubation with UCHL3, there is no accompanying increased signal of Rad51-GFP, further arguing that those Ub are not attached to Rad51.

We thank the reviewer for the valuable suggestion. We agree with the reviewer on this point. Indeed, it would be ideal to pull down RAD51-GFP, followed by another IP with an anti-ubiquitin antibody, before the *in vitro* deubiquitination assay was performed. However, we tried to do the experiments several times, but we failed to get enough ubiquitinated RAD51 for the further *in vitro* analysis. We reasoned that the amount of ubiquitinated RAD51 was relatively low, and the two consecutive IP experiments may cause more loss of the ubiquitinated RAD51. Therefore, we had to follow the deubiquitination assay in the previously published study (Luo, et al. 2016) to analyze how farrerol affected UCHL3 mediated deubiquitination on RAD51. Also, in the previously published study, using the same assay, Luo et al. demonstrated that mutating the ubiquitination sites on RAD51 would abolish the ubiquitination/deubiquitination process, indicating that this assay might be suitable for the study. Additionally, we repeated the experiment with the purified recombinant UCHL3-His from bacteria and obtained similar results.

Only shown in the point-by-point response letter

Fig. R-3

5. Fig. 2d-f shows beautifully that in the absence of UCHL3, farrerol has no effect on Rad51 foci. However, these experiments cannot exclude that farrerol acts on a factor that's downstream of UCHL3 in the HR pathway. To address this issue, the authors should rescue the UCHL3 knockout with either wildtype or non-binding mutant of UCHL3, such as R215A or K187A.

We thank the reviewer for the valuable suggestion. We examined RAD51 foci formation in irradiated UCHL3-KO cells with ectopically expressed WT UCHL3 or UCHL3 R215A/K187A mutant with or without farrerol treatment. As shown in new Fig.3g-h, WT UCHL3 with farrerol treatment, but not the UCHL3 R215A or K187A mutant, restored RAD51 foci formation. We have included the data in new Figure 3g-h.

Fig. 3g**Fig. 3h**
6. Given that farrerol enhances UChL3 activity about two-fold (Fig. 3h), one would expect that overexpression of UChL3 is sufficient to enhance HR. Has this been tested?

We thank the reviewer for the insightful suggestion. As the reviewer suggested, we performed HR assay using our well-established HR reporter cells. The result showed that UChL3 overexpression significantly enhanced the HR efficiency. We have included the data in new Supplementary Figure 2b.

Fig. S2b
7. The interpretation of Fig. 4i is problematic. Since UChL3 knockout did not provide any viable E19.5 pups by SCNT, one cannot conclude that farrerol acts through UChL3. There are many genes whose knockout could lead to the lethality phenotype and they are not farrerol targets.

We thank the reviewer for the suggestion. It has been reported that *Uchl3* knockout mice could survive to adulthood (Kurihara, et al. 2000). Meanwhile, we further demonstrated that *Uchl3* depletion did not influence the development of IVF embryos (Supplementary Fig. 8c). In addition, there was no significant difference in SCNT blastocyst rate between *Uchl3* depletion group (42.36±0.35%) and control group (40.69±7.67%). These results indicated that UChL3 itself has little effect on embryonic development. However, *Uchl3* depletion greatly attenuated farrerol-ameliorated developmental defects in SCNT embryos, including reduced pre-implantation development rate (Fig. 6b, c), increased γ H2AX foci (Fig. 6d), inadequate lineage differentiation (Fig. 6e-h), and reduced birth rate after implantation (Supplementary Table 4). Collectively, these results fully demonstrated that the function of farrerol in

improving SCNT efficiency is mainly dependent on *Uchl3*, that is, UCHL3 is the target of farrerol.

Fig. S8c

Fig. 6b

Fig. 6c

Fig. 6d

Fig. 6e

Fig. 6f

Fig. 6g

Fig. 6h

Table. S4

In vivo development of SCNT embryos				
mRNA injected	Treatment	No. of replicates	No. of 2-cell or 4-cell embryos transferred	No. of pups (% per ET)
/	Control-DMSO (1000×)	8	998	0.2±0.3
/	Farrerol (0.05 μM)	5	372	2.0±0.8***
/	Farrerol (0.1 μM)	4	152	4.0±2.5**
si/si Uchl3	Farrerol (0.05 μM)	3	173	0***
ET, embryo transfer.				
P<0.01; *P<0.001 as compared with the control group.				

8. The effects of farrerol on H3K27Me3 and transcription programs argue against the author's conclusion that UCHL3 is the main target of farrerol in SCNT. Given that farrerol has other well-documented activities outside HR, it seems more likely that farrerol is acting through multiple mechanisms in enhancing SCNT efficiency.

Thank the reviewers for the thought-provoking question. In the revised manuscript, we rewrote this description. Other potential targets of farrerol and other mechanisms apart from HR for enhancing reprogramming were discussed in the part of discussion.

In this study, by applying *in vitro* mass spectrometry assay and certain validation studies, we clearly identified UCHL3 is one of the main binding target of farrerol, which is essential for farrerol-mediated HR system. We found that SCNT embryos showed severely impaired HR repair, genomic instability and aneuploidy, whereas these defects could be largely rescued by farrerol treatment (Fig. 4b-f). However, *Uchl3* depletion greatly abolished farrerol-ameliorated developmental defects in SCNT embryos. For example, in *Uchl3*-deficient SCNT embryos, the number of γ H2AX foci, a marker of DNA damage, was not significantly reduced after farrerol treatment (Fig.6d). In addition, the pre-implantation and post-implantation development potentials of *Uchl3*-deficient farrerol-treated SCNT embryos were significantly reduced as compared to control farrerol-treated SCNT embryos (Fig. 6b; Supplementary Table 2 and 4).

Apart from the direct effect of farrerol on ameliorating DNA damage, we unexpectedly observed that certain epigenetic defects were partially rescued by farrerol. These improvements included an activation of H3K9me3-marked RRRs (Fig. 5a), an upregulation of ZGA-related genes (Fig. 5b), a more open chromatin state (Fig. 5g and Supplementary Fig. 6a) and an establishment of H3K27me3 in ICM of SCNT blastocysts (Fig. 5h). Thus, the effect of farrerol is truly not limited to HR pathway. Meanwhile, we demonstrated directly activating RAD51 through RS-1 did not have similar effects (Fig. 5h, 5i and Supplementary Fig. 6c-e; Supplementary Table 3). More interestingly, we were surprised to find that *Uchl3* depletion did abolish farrerol's effect on the establishment of H3K27me3 in ICM of SCNT embryos (Supplementary Table 3). This result indicated, other than HR repair, the other epigenetic changes caused by farrerol during SCNT process might also be dependent on UCHL3.

David A. Sinclair's team at Harvard Medical School has proposed a “RCM” (relocalization of chromatin modifiers) hypothesis, which suggested that DNA damage might cause repositioning of chromatin modifiers as to alter the epigenome landscape. More recently, they further confirmed this hypothesis in a paper published in *Cell*, and found that the repair of non-mutant DNA double-strand breaks leads to epigenomic changes, such as H3K27ac, H3K56ac, H3K27me3, and chromatin contacts (Yang, et al. 2023). This study helped us to understand and further confirmed a link between the DSB repair and the epigenome. Here, we show that farrerol makes multiple contributions to improving SCNT efficiency, which is an interesting topic for future research. In addition, whether breaking epigenetic barriers also influences the DNA repair pathway needs to be further investigated.

Fig. 5a

Expression level of RRRs (2-Cell, n=217)

Fig. 5b

Expression level of ZGA genes (2-Cell)

Fig. 5g

Fig. S6a

Fig. 5h

Table. S3

H3K27me3-positive-ICM blastocysts derived from IVF and SCNT embryos

Embryo type	mRNA injected	Treatment	No. of blastocysts examined	No. of H3K27me3-positive-ICM blastocysts	Ratio
NF blastocyst (Zhang et al.) ⁷⁰	N/A	N/A	34	34	100%
IVF blastocyst	/	DMSO (1000×)	15	15	100%
IVF blastocyst	/	Farrerol (0.05 μM)	17	17	100%
SCNT blastocyst (Zhang et al.) ⁷⁰	N/A	N/A	42	1	2.38%
SCNT blastocyst	/	DMSO (1000×)	14	1	7.14%
SCNT blastocyst	/	Farrerol (0.05 μM)	24	12	50%
SCNT blastocyst	/	RS-1 (10 μM)	13	2	15.38%
SCNT blastocyst	si/sgControl	Farrerol (0.05 μM)	16	7	43.75%
SCNT blastocyst	siUchl3 (20 μM)+sgUchl3 (50 ng/μL)	Farrerol (0.05 μM)	20	4	20%

IVF, *in vitro* fertilization; SCNT, somatic cell nuclear transfer; N/A, not applicable

70. Zhang M, Wang F, Kou Z, Zhang Y, Gao S. Defective chromatin structure in somatic cell cloned mouse embryos. *The Journal of biological chemistry* **284**, 24981-24987 (2009).

9. Line 426: “compared with the rough regulation pattern of RAD51”, please clarify what the authors mean by “rough regulation pattern”.

We thank the reviewer for the suggestion. We rewrote this part in the revised manuscript. We speculated that the observed differences between farrerol and RS-1 mediated changes in SCNT efficiency resulted from different mechanisms. RS-1 functions to promote the formation of active presynaptic filaments assembled by RAD51 and ssDNA (Jayathilaka, et al. 2008). Farrerol could promote the deubiquitination of RAD51 by specifically binding to UCHL3, which effectively promotes the RAD51-BRCA2 interaction (Luo, et al. 2016), and thus improves HR efficiency. Moreover, we demonstrated that the effects of farrerol are not limited to HR pathway. We found that farrerol could partially activate reprogramming-resistant regions (RRRs) (Fig. 5a), promote the establishment of H3K27me3 modification (Fig. 5h), and enhance the expression of lineage differentiation genes (Supplementary Fig. 6f). In the revised Discussion part, we discussed the different regulatory mechanisms between farrerol and RS-1. We have included the data in new Figure 5a, 5h and Supplementary Figure 6f.

Fig. 5a

Fig. 5h

Fig. S6f

Reviewer #3 (Remarks to the Author):

The paper by W. Zhang et al. reports that farrerol, a natural flavanone, directly targets a deubiquitinase UCHL3, which activates a recombinase RAD51, thus promoting homologous recombination (HR) DNA repair. The authors' precise LiP-SMap combined with a BIAcore assay as well as cocrystallization experiments identified direct binding of farrerol to UCHL3 through specific amino acid residues. Finally, the authors demonstrated that the birth rate of SCNT was significantly improved from < 1% to about 4% by treatment with farrerol. RNA-seq analysis revealed that this improvement could be attributable to corrected expression profiles of developmentally important genes and repression of somatic cell-related genes. The experiments were well-designed, the results were clearly presented, and final conclusions were logically drawn from the experimental results obtained. Overall, the paper is of a highly scientific quality, and will attract broad interests of researchers in the cell biology as well as developmental /reproductive biology. However, there are some points to be improved so that the paper can be a more convincing paper in the related fields.

We are delighted by the reviewer's positive comments on our manuscript.

Major points

1. The most striking finding in this study is the advantageous changes in the transcriptomic feature of SCNT embryos following farrerol treatment. As far as its chemical and biochemical backgrounds are concerned, farrerol is involved in DNA repair at the time of genomic reprogramming, not in genomic (epigenetic) reprogramming per se. Although the authors mentioned that it is an interesting topic for future research, I strongly recommend the authors to perform additional experiments that may help the authors propose some possible mechanism, because this could be the central notion of this study. For example, if the farrerol treatment induces open chromatin in reconstructed oocytes, this might offer a reasonable scenario for efficient SCNT (like TSA treatment). ATAC-seq or similar analysis may work well. Alternative ways may include observation of the level of maternal H3.3 incorporation into the donor nucleus in the reconstructed oocytes, if the authors easily prepare H3.3 mRNA for injection.

We thank the reviewer for the insightful suggestion. HA-H3.3 mRNA was prepared and injected into oocytes for SCNT and IVF experiments. By immunofluorescence staining with an anti-HA antibody, we found that the farrerol-treated group showed higher HA intensity compared with the control group (Fig. 5g and Supplementary Fig. 6a). This suggests that farrerol could facilitate the incorporation of histone H3.3 into the nucleus, thus promoting an open chromatin state. Consistently, partial activation of H3K9me3-marked RRRs (Fig. 5a) and restored expression of ZGA-related genes (Fig. 5b) were observed after farrerol treatment. Our results indicated a link between HR repair and the epigenome, which is consistent with the "RCM" hypothesis by David A. Sinclair. More recently, Sinclair group demonstrated that the repair of non-mutated DNA double-strand breaks would lead to

epigenomic changes involving H3K27ac, H3K56ac, H3K27me3, and chromatin contacts (Yang, et al. 2023). However, the underlying mechanism between DSB repair and specific epigenetic modification remains unclear and needs to be further explored. We have included the data in new Figure 5a-b, 5g and Supplementary Figure 6a.

Fig. 5g

Fig. S6a

Fig. 5a

Fig. 5b

2. Another intriguing finding is the lack of the positive effect of RS-1 (RAD51 activator) treatment on SCNT, although farrerol promotes RAD51-mediated DNA repair as well. Lee et al. (ref. 22) reported that the mouse cloning efficiency (birth rate of cloned mice) increased from 0.4% to 5.2% by RS-1 treatment. In vitro development of SCNT embryos was also significantly improved; from 35% to 66% blastocyst rate. In this study, the blastocyst rate was about 40% following RS-1 treatment, showing no improvement from the control SCNT level. By what reason do the authors assume these two different outcomes occurred? Why RS-1 did not have any positive effect on SCNT in this study? In the discussion, the authors mentioned that “farrerol functions in a more refined way”, but this is not a scientific explanation.

We thank the reviewer for raising these issues. We made changes in discussing the farrerol and RS-1 mediated changes in SCNT efficiency in the revised manuscript. In the previous version, we found that the pre-implantation developmental rate of SCNT embryos treated with RS-1 was similar to that of the control by three independent experiments. In this revised version, we further conducted a test experiment according to the concentration gradient given in the Lee’s article (Lee, et al. 2021). The results showed that the blastocyst rate of RS-1 at 5 μ M was $33.63 \pm 7.54\%$, at 10 μ M was $46.67 \pm 12.78\%$, at 15 μ M was $29.38 \pm 8.55\%$, and that of the control group was

37.69±7.62% (Supplementary Table 2 and Fig. R-4). Although consistent with Lee's article that 10 μM was the optimal concentration, we found that the developmental rate was lower than their report (75±4.2%).

The different observations between Lee's group and our study might be explained by that the treatment system in Lee's article was a little bit different from ours. They started the treatment at the time of activation and lasted for 22 hours, while we started treatment 6 hours after activation and treated for 16 hours. Therefore, we carried out experiments according to their treatment system and found that the SCNT blastocyst rate could reach 63.58±0.42% at the concentration of 10 μM of RS-1, which was still lower than 70.02±10% of farrerol group in our system (Supplementary Table 2). In addition to the direct effect of farrerol on HR, we further demonstrated that farrerol might play other roles, such as in the restoration of H3K27me3-positive-ICM and the enhanced lineage differentiation, which were not evident in the RS-1 treated group (Fig. 5h and Supplementary Figure. 6c-e). Collectively, we showed a better effect of farrerol than RS-1 on promoting SCNT efficiency.

Table. S2

Preimplantation Development of IVF and SCNT embryos							
Embryo type	mRNA injection	Treatment	No. of replicates	No. of reconstructed 2-cell embryos	%4-cell per 2-cell ±SD	%morula per 2-cell ±SD	%blastocyst per 2-cell ±SD
IVF	/	Control-DMSO (1000×, maintain)	3	60	100±0.00	96.11±2.83	93.44±1.23
	/	Mirin (50 μM, maintain)	4	58	0	0	0
	/	Mirin (50 μM, 16 h)	2	45	96.15±3.85	92.31±7.69	81.78±2.83*
	/	Farrerol (0.05 μM, maintain)	3	65	100±0.00	98.25±2.48	92.45±1.62
	/	Farrerol (0.05 μM, 16 h)	2	40	100±0.00	100±0.00	100±0.00**
SCNT	/	Control-DMSO (1000×, maintain or 16 h)	16	346	70.16±7.28	50.80±9.19	37.69±7.62
	/	Mirin (50 μM, maintain)	4	0	0	0	0
	/	Mirin (50 μM, 16 h)	3	53	23.21±11.04***	13.84±9.00***	7.14±6.73***
	/	Farrerol (0.05 μM, maintain)	2	38	63.06±1.94	0	0
	/	Farrerol (0.05 μM, 16 h)	10	217	89.95±5.20***	81.09±10.80***	70.02±10.00***
	/	Farrerol (0.1 μM, 16 h)	6	148	85.83±6.29***	74.50±7.57***	53.86±7.78***
	/	Farrerol (0.2 μM, 16 h)	3	58	41.55±2.62***	39.79±3.33	32.92±3.39
	/	RS-1 (5 μM, 16 h)	2	57	72.95±5.31	52.56±0.38	33.63±7.54
	/	RS-1 (10 μM, 16 h)	5	117	80.53±10.37*	67.61±16.87*	46.67±12.78
	/	RS-1 (10 μM, 22 h)	2	44	91.37±3.37**	77.47±1.47**	63.58±0.42***
IVF	si/gControl	Control-DMSO (1000×, 16 h)	2	60	100±0.00	100±0.00	98.28±1.72
	siUchl3 (20 μM)+sgUchl3 (50 ng/μL)	DMSO (1000×, 16 h)	2	56	98.21±1.79	96.43±0.00	91.07±1.79
SCNT	si/gControl	Control-DMSO (1000×, 16 h)	3	59	69.59±2.98	50.79±6.54	42.36±0.35
	siUchl3 (20 μM)+sgUchl3 (50 ng/μL)	DMSO (1000×, 16 h)	3	58	65.33±6.23	41.92±8.43	40.69±7.67
	si/gControl	Farrerol (0.05 μM, 16 h)	2	39	90.08±4.37*	80.16±8.73*	69.44±2.78***
	siUchl3 (20 μM)+sgUchl3 (50 ng/μL)	Farrerol (0.05 μM, 16 h)	4	89	70.00±5.11	63.34±5.37	52.78±2.78**

IVF, *in vitro* fertilization; SCNT, somatic cell nuclear transfer
*P<0.05; **P<0.01; ***P<0.001 as compared with the corresponding control group.

Only shown in the point-by-point response letter

Fig. R-4

Fig. 5h**Fig. S6e**
3. The authors argued that placental hyperplasia associated with mouse SCNT was alleviated by farrerol treatment. At least three SCNT papers so far have reported that loss of maternal noncanonical imprinting was responsible for this placental hyperplasia. As the authors produced cloned mice with PWKxB6 donors, the allelic data on the noncanonical imprinting status of the placentas of farrerol-treated SCNT should be presented.

We thank the reviewer for the insightful suggestion. For detecting the reported non-canonical imprinted genes related to placental hyperplasia (Inoue, et al. 2020; Wang, et al. 2020; Xie, et al. 2022), transcriptome sequencing was performed on E7.5 extraembryonic ectoderm (ExE) of SCNT and IVF embryos. The results showed that the expression of a number of non-canonical imprinted genes such as *Slc38a4* and *Sfmbt2*, that were abnormally overexpressed in SCNT ExE, was reduced after farrerol treatment (Supplementary Fig. 7e). However, although these non-canonical imprinted genes showed reduced expression, they still remained a parental expression pattern, which differs from the paternally dominant expression pattern in the IVF group (Fig. R-5). In addition, we observed the restored expression of certain canonical genes (such as *H19* and *Igf2*) (Supplementary Fig. 7f), which indicated multiple dimensions of epigenome repairment by farrerol. As a result, the placental weight of farrerol-treated group (0.277 g) was not significantly different from that of *Slc38a4*-maternal knockout group (0.248 g) (Supplementary Fig. 7c). We have included the data in new Supplementary Figure 7c, 7e, 7f.

Fig. S7e

Fig. S7f

Only shown in the point-by-point response letter

Fig. R-5

Fig. S7c

4. As shown in Fig. 4d and Extended Data Fig. 3b, farrerol exerted its effect on the 2-cell to 4-cell transition, indicating that farrerol improved zygotic genome activation. However, the authors did not present any transcriptome data that support this. It is

important to see if the farrerol treatment can activate the reprogramming resistant regions (RRRs) repressed by H3K9me3 in SCNT embryos (Matoba et al. Cell 2014).

We thank the reviewer for the insightful suggestion. We analyzed the H3K9me3-marked reprogramming resistance regions (RRRs) in IVF-embryos and SCNT-embryos with and without farrerol treatment (Matoba, et al. 2014). The result showed that the RRRs was partially activated in farrerol-treated SCNT embryos (Fig. 5a). In addition, certain ZGA-related genes such as *Dux* and *Zscan4c* were also activated (Fig. 5b). We have included the data in new Figure 5a, b.

Minor points

1. Abstract: In the last sentence, there is: “provided a more feasible method to promote SCNT efficiency when compared to other methods”. I wonder what data in this study guarantee “more feasible than other methods”. The authors would better explain this.

We thank the reviewer for the question. In this study, we demonstrated that the farrerol treatment during SCNT process could enhance HR repair, transcriptional and epigenetic network and promote the development of SCNT embryos. Compared with other methods that require genetic and epigenetic manipulations (Gao, et al. 2018; Inoue, et al. 2020; Matoba, et al. 2014; Wang, et al. 2020; Xie, et al. 2022; Yang, et al. 2021), supplementing farrerol in the culture medium is relatively convenient. Moreover, the mechanistic differences in promotion of SCNT efficiency between farrerol and RS-1 are also explained in the discussion section of the revised version.

2. Line 104: ref. 18 is not a SCNT paper.

We thank the reviewer for pointing out the reference problem. We have removed this reference in the revised version.

3. Lines 111-113: “As a comparison...”. Appropriate reference(s) should be cited here.

We thank the reviewer for pointing it out. We have added appropriate reference in the revised version.

4. Line 275: Fig. 4d -> Fig. 4d-e

We have corrected the error.

5. Line 275: Extended Data Fig. 3a shows IVF embryo development, not SCNT.

We have made the correction.

6. Line 363: The experimental conditions of *Uchl3*-sg/siRNA were different between SCNT and IVF; oocytes were used for the former, but zygotes were used for the latter. It is essential to inject sg/siRNA into oocytes before IVF to see the effect of sg/siRNA on normal embryonic development more correctly.

We thank the reviewer for the suggestion. As suggested by the reviewer, we conducted the injection of sg/siRNA into MII oocytes before IVF. As shown in new Supplementary Fig. 8c, we did not observe any changes in the morphology of embryos in *Uchl3*-depleted IVF group, and the embryo development was not affected. We have included the data in new Supplementary Figure 8c.

Fig. S8c

7. Line 367: Fig. 4j -> Fig. 4i-j

We thank the reviewer for pointing it out. We corrected this error.

8. Discussion: In the Discussion section, it may be more natural to discuss first about UCHL3 as a target of farrerol, in line with the order of the Results section. For example, the paragraph “Previous studies have revealed that farrerol exhibits...” (Line 465) can be the first.

We thank the reviewer for pointing it out. We have adjusted the order of the discussion sections.

Reference

- Gao, R., et al.
2018 Inhibition of Aberrant DNA Re-methylation Improves Post-implantation Development of Somatic Cell Nuclear Transfer Embryos. *Cell Stem Cell* 23(3):426-435 e5.
- Inoue, K., et al.
2020 Loss of H3K27me3 imprinting in the Sfmt2 miRNA cluster causes enlargement of cloned mouse placentas. *Nat Commun* 11(1):2150.
- Jayathilaka, K., et al.
2008 A chemical compound that stimulates the human homologous recombination protein RAD51. *Proc Natl Acad Sci U S A* 105(41):15848-53.
- Kurihara, L. J., et al.
2000 Expression and functional analysis of Uch-L3 during mouse development. *Mol Cell Biol* 20(7):2498-504.
- Lee, A. R., et al.
2021 Genome stabilization by RAD51-stimulatory compound 1 enhances efficiency of somatic cell nuclear transfer-mediated reprogramming and full-term development of cloned mouse embryos. *Cell Prolif* 54(7):e13059.
- Luo, K., et al.
2016 A phosphorylation-deubiquitination cascade regulates the BRCA2-RAD51 axis in homologous recombination. *Genes Dev* 30(23):2581-2595.
- Matoba, S., et al.
2014 Embryonic development following somatic cell nuclear transfer impeded by persisting histone methylation. *Cell* 159(4):884-95.
- Matoba, S., et al.
2018 Loss of H3K27me3 Imprinting in Somatic Cell Nuclear Transfer Embryos Disrupts Post-Implantation Development. *Cell Stem Cell* 23(3):343-354 e5.
- Wang, L. Y., et al.
2020 Overcoming Intrinsic H3K27me3 Imprinting Barriers Improves Post-implantation Development after Somatic Cell Nuclear Transfer. *Cell Stem Cell* 27(2):315-325 e5.
- Xie, Z., W. Zhang, and Y. Zhang
2022 Loss of Slc38a4 imprinting is a major cause of mouse placenta hyperplasia in somatic cell nuclear transferred embryos at late gestation. *Cell Rep* 38(8):110407.
- Yang, G., et al.
2021 Dux-Mediated Corrections of Aberrant H3K9ac during 2-Cell Genome Activation Optimize Efficiency of Somatic Cell Nuclear Transfer. *Cell Stem Cell* 28(1):150-163 e5.
- Yang, J. H., et al.
2023 Loss of epigenetic information as a cause of mammalian aging. *Cell*.
- Zhang, C., et al.
2022 US-align: universal structure alignments of proteins, nucleic acids, and macromolecular complexes. *Nat Methods* 19(9):1109-1115.

REVIEWERS' COMMENTS

Reviewer #1 (Remarks to the Author):

The authors addressed the reviewers comments very well, with a few exceptions that still need to be addressed because they incurred technical issues, but which should be solvable.

Rad51 intensity instead of foci. It should be possible to wash out nucleocytoplasmic Rad51 to bring out foci and count those. Intensity of nucleoplasmic Rad51 is not informative.

Reviewer #2 (Remarks to the Author):

In the revised manuscript, the authors have addressed the majority of my concerns. However, inconsistencies in the description of how farrerol activates UCHL3 needs to be clarified.

On page 5, the authors stated "farrerol occupied the ubiquitin-interacting pocket of UCHL3" and "The alignment indicated that farrerol did not affect ubiquitin binding by UCHL3". These two sentences seem to be contradictory to each other. On page 6, the authors stated "The cocrystal structure of the UCHL3-farrerol complex revealed that farrerol might influence the catalytic activity of UCHL3 by directly anchoring in the ubiquitin- interacting pocket of UCHL3". If farrerol does not affect ubiquitin binding, how does it affect the catalytic activity by anchoring in the ubiquitin binding pocket?

Reviewer #3 (Remarks to the Author):

The authors revision has perfectly answered to my questions/comments and now I have no further comment to the paper.

Responses to reviewer's comment

Reviewer #1 (Remarks to the Author):

The authors addressed the reviewers comments very well, with a few exceptions that still need to be addressed because they incurred technical issues, but which should be solvable.

Rad51 intensity instead of foci. It should be possible to wash out nucleocytoplasmic Rad51 to bring out foci and count those. Intensity of nucleoplasmic Rad51 is not informative.

We thank the reviewer for the valuable suggestion. As reviewer suggested, we performed immunofluorescence experiments and washed out nucleocytoplasmic RAD51 to bring out foci with cytoskeleton (CSK) buffer (10 mM PIPES, 300 mM Sucrose, 100 mM NaCl, 3 mM MgCl₂, 1 mM EGTA), and counted the number of RAD51 foci in embryos at the cleavage stage. In consistence with previous data, we found that the number of RAD51 foci in SCNT embryos was fewer than that in IVF embryos, and the farrerol treatment greatly increased the number of RAD51 foci in SCNT embryos compared to that in DMSO-treated SCNT embryos. We have included these data in new Figure 4f and Supplementary Figure 4f.

Reviewer #2 (Remarks to the Author):

In the revised manuscript, the authors have addressed the majority of my concerns. However, inconsistencies in the description of how farrerol activates UCHL3 needs to be clarified.

On page 5, the authors stated "farrerol occupied the ubiquitin-interacting pocket of UCHL3" and "The alignment indicated that farrerol did not affect ubiquitin binding by UCHL3". These two sentences seem to be contradictory to each other. On page 6, the authors stated "The cocystal structure of the UCHL3-farrerol complex revealed that farrerol might influence the catalytic activity of UCHL3 by directly anchoring in

the ubiquitin- interacting pocket of UCHL3". If farrerol does not affect ubiquitin binding, how does it affect the catalytic activity by anchoring in the ubiquitin binding pocket?

We thank the reviewer for pointing out the issues and sorry for the confusion. The description that "farrerol occupied the ubiquitin-interacting pocket of UCHL3" is not appropriate. An overlay of the structures of farrerol-UCHL3 complex and UCHL3 in complex with ubiquitin (1XD3) showed that farrerol did not induce extensive conformational changes. The ubiquitin-AMC assay revealed that farrerol affected K_m in a relatively mild way (from 47.72 to 56.27 nM), while it greatly increased K_{cat} from 11.99 to 19.64 s⁻¹ (Fig. 3i), indicating that farrerol-mediated activation of UCHL3 enzymatic activity is probably not through stimulating the UCHL3-ubiquitin association, or at most the activation is only partially dependent on the increase in UCHL3-ubiquitin association. Moreover, we propose that farrerol may act as an allosteric activator of UCHL3 as the predicted allosteric sites in UCHL3 using the software of Allositepro (Song et al., *J Chem Inf Model*, 2017) overlap with the binding sites of farrerol (a). Also, after interacting with farrerol, three residues (residues 56-58) of UCHL3 were pushed away by 1.7 Å (b). This conformational change may influence the catalytic activity of UCHL3. We have modified this part in the revised vision.

Only shown in the point-by-point response letter

a. Left, overall structure with the surface of UCHL3 colored in pink, and predicted allosteric site colored in blue/white. Inset, magnified view of the allosteric site of UCHL3. UCHL3 is displayed as a pink cartoon, and farrerol is shown in sticks. The residues within the allosteric site of UCHL3 are highlighted within the stick representation. Hydrogen bonds are indicated by yellow dotted lines. **b.** The overlay of UCHL3 in complex with Ub-VME (1XD3, colored in cyan) with farrerol-UCHL3 complex (colored in magentas) was showed. F56, P57, and I58 of 1XD3 are colored in red, while these residues of farrerol-UCHL3 complex are colored in yellow.

Reviewer #3 (Remarks to the Author):

The authors revision has perfectly answered to my questions/comments and now I have no further comment to the paper.

We thank the reviewer for very positive evaluation of our revised manuscript.